# Na$_V$1.1 is essential for proprioceptive signaling and motor behaviors

Cyrrus M Espino[1], Cheyanne M Lewis[1], Serena Ortiz[2], Miloni S Dalal[3], Snigdha Garlapalli[4], Kaylee M Wells[5], Darik A O'Neil[5], Katherine A Wilkinson[2], Theanne N Griffith[1,5]*

[1]Department of Physiology and Membrane Biology, University of California, Davis, Davis, United States; [2]Department of Biological Science, San Jose State University, San Jose, United States; [3]Department of Pharmacology, Physiology, and Neuroscience, New Jersey Medical School, Rutgers University, Newark, United States; [4]Undergraduate Program in Psychology, University of California, Davis, Davis, United States; [5]Neurobiology Course, Marine Biological Laboratory, Woods Hole, United States

*For correspondence:
tgriffith@ucdavis.edu

Competing interest: The authors declare that no competing interests exist.

**Abstract** The voltage-gated sodium channel (Na$_V$), Na$_V$1.1, is well-studied in the central nervous system; conversely, its contribution to peripheral sensory neuron function is more enigmatic. Here, we identify a new role for Na$_V$1.1 in mammalian proprioception. RNAscope analysis and in vitro patch-clamp recordings in genetically identified mouse proprioceptors show ubiquitous channel expression and significant contributions to intrinsic excitability. Notably, genetic deletion of Na$_V$1.1 in sensory neurons caused profound and visible motor coordination deficits in conditional knockout mice of both sexes, similar to conditional Piezo2-knockout animals, suggesting that this channel is a major contributor to sensory proprioceptive transmission. Ex vivo muscle afferent recordings from conditional knockout mice found that loss of Na$_V$1.1 leads to inconsistent and unreliable proprioceptor firing characterized by action potential failures during static muscle stretch; conversely, afferent responses to dynamic vibrations were unaffected. This suggests that while a combination of Piezo2 and other Na$_V$ isoforms is sufficient to elicit activity in response to transient stimuli, Na$_V$1.1 is required for transmission of receptor potentials generated during sustained muscle stretch. Impressively, recordings from afferents of heterozygous conditional knockout animals were similarly impaired, and heterozygous conditional knockout mice also exhibited motor behavioral deficits. Thus, Na$_V$1.1 haploinsufficiency in sensory neurons impairs both proprioceptor function and motor behaviors. Importantly, human patients harboring Na$_V$1.1 loss-of-function mutations often present with motor delays and ataxia; therefore, our data suggest that sensory neuron dysfunction contributes to the clinical manifestations of neurological disorders in which Na$_V$1.1 function is compromised. Collectively, we present the first evidence that Na$_V$1.1 is essential for mammalian proprioceptive signaling and behaviors.

## Editor's evaluation

This article provides insight into the importance of a voltage-gated sodium channel in proprioceptors, a group of mechanosensory neurons that target muscle. Using pharmacology, gene knockout, behavior, and histology in mice, the authors show quite convincingly that Na$_V$1.1 in sensory neurons is essential for normal motor behavior and contributes to proprioceptor excitability. The work has interesting implications for human subjects with inherited variants of Na$_V$1.1.

## Introduction

Voltage-gated sodium channels (Na$_V$s) are critical mediators of neuronal excitability and are responsible for action potential generation and propagation (*Ahern et al., 2016*; *Bean, 2007*). In the mammalian nervous system, there are nine isoforms (Na$_V$1.1–1.9), each with unique biophysical properties, as well as distinguishing cellular expression and subcellular localization patterns (*Bennett et al., 2019*; *Catterall, 2017*). Of these different subtypes, Na$_V$1.1 is notable for its role in brain disease (*Escayg and Goldin, 2010*; *Mulley et al., 2005*; *Ogiwara et al., 2007*). Indeed, *Scn1a*, the gene that encodes Na$_V$1.1, is referred to as a 'super culprit' gene, with over 1000 associated mutations that lead to abnormal brain function, resulting in brain disorders such as epilepsy and migraine, as well as neurodivergent phenotypes, such as autism spectrum disorder (*Ding et al., 2021*; *Lossin, 2009*). Homozygous *Scn1a$^{-/-}$* global knockout mice are ataxic and die by postnatal day (P) 15, while heterozygous *Scn1a$^{+/-}$* animals develop seizures and begin to die sporadically starting at P21 (*Yu et al., 2006*). In addition to the central nervous system (CNS), Na$_V$1.1 is also expressed in the peripheral nervous system (PNS) (*Sharma et al., 2020*; *Usoskin et al., 2015*); yet, the prominent role this channel plays in brain function has left its physiological roles in sensory neuron populations understudied.

Peripheral sensory neurons of the dorsal root ganglia (DRG) and trigeminal ganglia (TG) are tasked with encoding somatic sensations, such as touch, temperature, pain, and proprioception, and are anatomically and functionally heterogenous (*Kupari et al., 2021*; *Nguyen et al., 2021*; *Oliver et al., 2021*; *Wu et al., 2021*). *Scn1a* transcript and protein have been observed primarily in myelinated mechanosensory DRG and TG neurons (*Fukuoka et al., 2008*; *Ho and O'Leary, 2011*; *Osteen et al., 2016*). Indeed, subcutaneous injection of the Na$_V$1.1 activator, Hm1a, into mouse hind paw causes noninflammatory mechanical pain and spontaneous pain behaviors (*Osteen et al., 2016*). Interestingly, pharmacological inhibition of Na$_V$1.1 does not affect mechanical thresholds in uninjured mice but does reduce mechanical pain in a spared-nerve injury model (*Salvatierra et al., 2018*), suggesting that Na$_V$1.1 may have a more prominent role in mechanical pain as opposed to normal touch sensing. Na$_V$1.1 in TG neurons has also been reported to mediate mechanical pain in an orbitofacial chronic constriction injury model (*Pineda-Farias et al., 2021*). In addition to somatosensory neurons, Na$_V$1.1 is found in colon-innervating vagal neurons, where it contributes to firing of colonic mechanonociceptors and is upregulated in a mouse model of chronic visceral hypersensitivity (*Osteen et al., 2016*; *Salvatierra et al., 2018*). Lastly, Na$_V$1.1 contributes to action potential firing in a subset of DRG neurons that express the cold-sensitive ion channel, transient receptor potential melastin 8 (TRPM8), suggesting that the channel may also contribute to thermosensory transmission (*Griffith et al., 2019*). While most data support a role for Na$_V$1.1 in pain, the limited number of studies that have investigated Na$_V$1.1 function in sensory neurons has left gaps in our knowledge regarding other potential roles this channel may play in somatosensation.

Given the relatively underexplored role of Na$_V$1.1 in the PNS, we set out to determine what other somatosensory modalities rely on Na$_V$1.1 expression in sensory neurons. Here, we show that 100% of proprioceptors express *Scn1a* mRNA, where it makes notable contributions to the somal whole-cell sodium current and intrinsic excitability. A functional role for Na$_V$1.1 in proprioceptive signaling was also supported by ex vivo electrophysiological recordings from functionally identified muscle spindle afferents. Importantly, mice lacking Na$_V$1.1 in all sensory neurons display visible and profound motor deficits and ataxic-like behavior, which were quantified in rotarod and open-field assays. Surprisingly, we found that Na$_V$1.1 is haploinsufficient for normal proprioceptor function and behavior, in ex vivo recordings and the open-field assay, respectively. Collectively, our data provide the first evidence that peripherally expressed Na$_V$1.1 is critical for sensory proprioceptive signaling and motor coordination.

## Results

Most studies have localized Na$_V$1.1 expression primarily to myelinated sensory neurons that transmit mechanical signals (*Fukuoka et al., 2008*; *Ho and O'Leary, 2011*; *Osteen et al., 2016*; *Wang et al., 2011*). In line with prior work, RNAscope analysis of DRG sections from adult mice showed that 93% of myelinated neurons, as determined by neurofilament heavy chain (NFH) labeling, express *Scn1a* transcripts (*Figure 1A*). RNA-sequencing datasets have consistently identified Na$_V$1.1 expression in proprioceptors; thus, we next analyzed Na$_V$1.1 expression in these cells using a Parvalbumin$^{Cre}$;Rosa26$^{Ai14}$

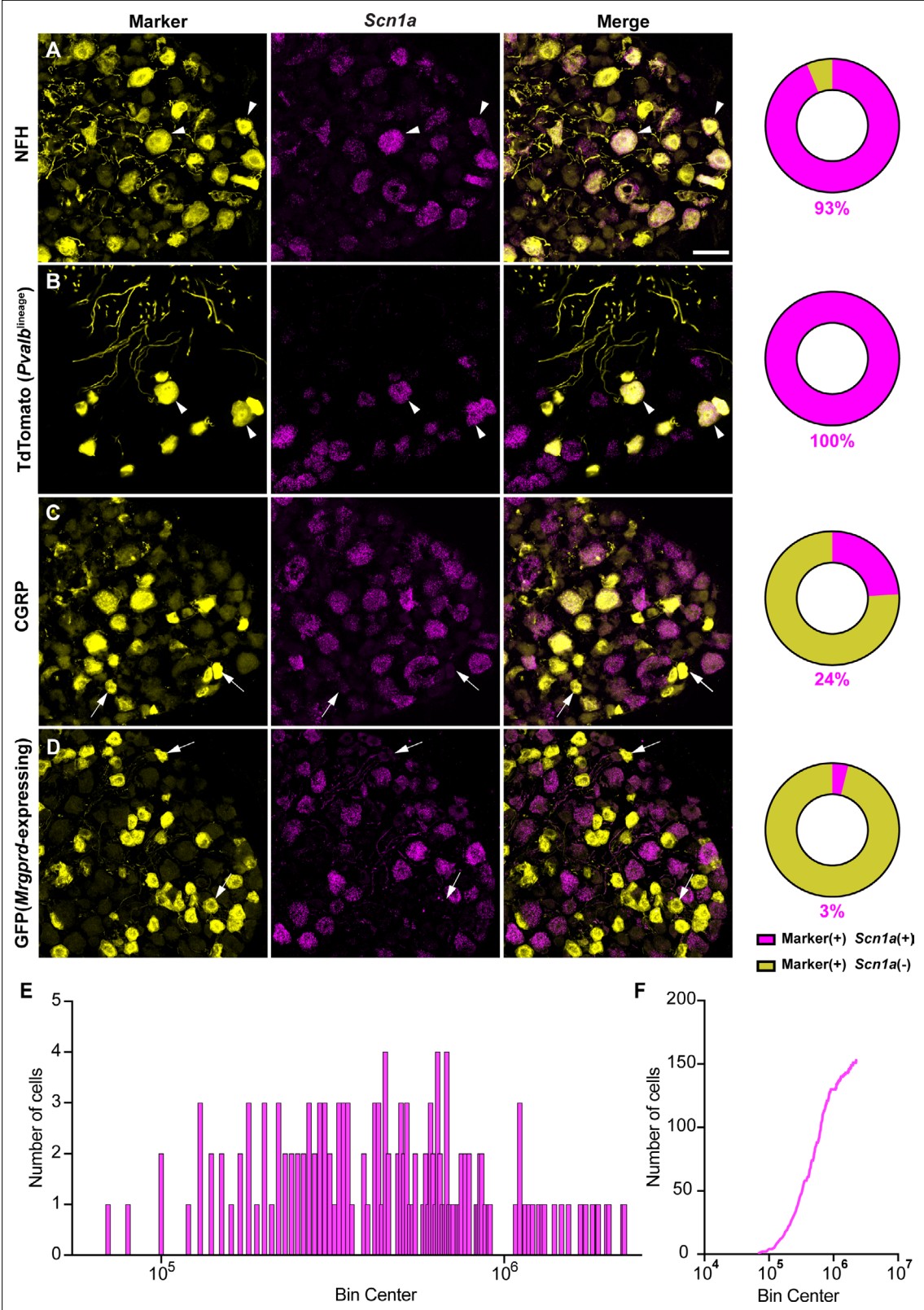

**Figure 1.** Na$_\text{v}$1.1 is ubiquitously expressed in genetically identified proprioceptors. (**A–D**) Representative confocal images of cryoprotected adult dorsal root ganglia (DRG) sections (25 μm) with pie chart quantifications indicating the percentage of *Scn1a+* and *Scn1a-* neurons in each subpopulation (magenta and yellow, respectively). Images were acquired with a ×40, 0.9 NA water-immersion objective. Sections were hybridized using RNAscope with probes targeting *Scn1a* (*Scn1a*, magenta) and stained with the following antibodies (yellow): (**A**) anti-neurofilament heavy (NFH, n = 787), (**B**) anti-DsRed

*Figure 1 continued on next page*

to label TdTomato+ proprioceptors (n = 153), (**C**) anti-calcitonin gene-related peptide (CGRP, n = 877), and (**D**) anti-GFP to label Mrgprd+ neurons (n = 744). DRG from C57BL/6, *Pvalb*^Cre;^*Rosa26*^Ai14^, and *Mrgprd*^GFP^ mice of both sexes were used. Scale bar 50 μm. White arrowheads indicate *Scn1a*+ neurons while white arrows indicate *Scn1a*- neurons. Frequency (**E**) and cumulative (**F**) distribution plots of integrated fluorescence density of the *Scn1a* signal in TdTomato+ proprioceptors (n = 153). n = cells.

reporter line (*Pvalb*^Ai14^) and found that 100% of genetically identified proprioceptors were positive for *Scn1a* message (*Figure 1B*). This contrasted with low expression of *Scn1a* mRNA in both calcitonin gene-related peptide (CGRP)-expressing neurons, which represent peptidergic nociceptors, and non-peptidergic polymodal *Mrgprd*-expressing nociceptors (24 and 3%, respectively, *Figure 1C and D*). Frequency and cumulative distribution plots show the spread of integrated fluorescence density measurements obtained for *Scn1a* transcripts in proprioceptors (*Figure 1E and F*).

In addition to Na$_V$1.1, Na$_V$1.6 and Na$_V$1.7 expression has also been reported in proprioceptors (*Carrasco et al., 2017*). As with *Scn1a*, mRNA for *Scn8a* and *Scn9a* is also found in 100% of genetically identified proprioceptors (*Figure 2A and B*). Cumulative distribution plots of *Scn8a* and *Scn9a* integrated fluorescence density measurements showed higher variability as compared to Na$_V$1.1 (*Figures 1E and 2C and D*). This was quantified using the coefficient of variation (CV), a relative measure of the extent of variations within data. The CV for *Scn1a* transcript expression was calculated to be 75.6, whereas this value increased to 97.3 and 88.1 for *Scn8a* and *Scn9a*, respectively. This indicates that while all three isoforms are ubiquitously expressed in proprioceptors, the relative levels differ, with Na$_V$1.1 having the most consistent level of expression across neurons analyzed. Furthermore, the average integrated density of the Na$_V$1.1 signal for a given proprioceptive DRG neuron was significantly higher than the average integrated densities calculated for both Na$_V$1.6 and Na$_V$1.7 (*Figure 2E*, p<0.0001).

Due to the ubiquitous expression of Na$_V$1.1, Na$_V$1.6, and Na$_V$1.7 in proprioceptors, we sought to determine the relative contributions of these isoforms to the proprioceptor whole-cell sodium current (I$_{Na}$). We performed in vitro voltage-clamp experiments on TdTomato+ neurons harvested from thoracic spinal levels of adult *Pvalb*^Ai14^ mice (*de Nooij et al., 2013*) and used serial application of selective Na$_V$ channel antagonists to determine the specific contributions of Na$_V$1.1, Na$_V$1.6, and Na$_V$1.7 (*Figure 2F*). We first applied the selective Na$_V$1.1 antagonist ICA 1214314 (ICA, 500 nM), followed by 9-anhydroustetrodoxin (AH-TTX, 300 nM), which is a selective Na$_V$1.6 blocker but also partially blocks Na$_V$1.1 (*Denomme et al., 2020*; *Griffith et al., 2019*). We reasoned that by first blocking the Na$_V$1.1-mediated current, the effect of AH-TTX should largely be due to inhibition of Na$_V$1.6. Finally, we blocked Na$_V$1.7 channels using the antagonist PF-05089771 (25 nM), followed by tetrodotoxin (TTX, 300 nM) to block any residual current, as proprioceptors do not express TTX-resistant Na$_V$s. On average, 7.8% of the current remained unblocked following serial application of these antagonists due to incomplete block by the drugs used. We found that the ICA-sensitive component was 44.8% of I$_{Na}$. Conversely, 14.5 and 32.9% of I$_{Na}$ was sensitive to AH-TTX and PF-05089771, respectively (*Figure 2G*). No significant effect of the 0.1% DMSO vehicle solution on I$_{Na}$ amplitude was observed (*Figure 2— figure supplement 1*). Collectively, these data suggest that in proprioceptors Na$_V$1.1 is a dominant functional Na$_V$ subtype.

We next determined the biophysical features of the whole-cell sodium current (I$_{Na}$) in proprioceptors, which has not been previously reported (*Figure 3A–D*). The current–voltage relationship shows the first detectable current appeared at voltages near –50 mV and was maximal at voltages near –30 mV when evoked from a holding potential of –90 mV (*Figure 3B*). Voltage dependence of peak conductance was best fit to a single Boltzmann function and the voltage for half-maximal activation was –38.7 mV (*Figure 3C*). The voltage dependence of inactivation was determined with 40 ms prepulse steps ranging from –120 mV to +10 mV. The midpoint of the inactivation curve was –64.5 mV and was best fit to a single Boltzmann function (*Figure 3D*). To analyze recovery from fast inactivation, TdTomato+ neurons were depolarized to –20 mV, followed by a series of recovery periods ranging from 0.5 ms to 10 ms before a second test step to –20 mV was given to assess sodium channel availability. I$_{Na}$ recovery was rapid ($\tau$ = 0.54ms), with greater than 50% of I$_{Na}$ recovered after 0.5 ms (*Figure 3E*). Finally, entry into slow inactivation was determined. Cells were held at 0 mV during conditioning voltage steps ranging from 10 ms to 2000 ms, separated by two 2 ms pulses to –20 mV to compare channel availability before and after the conditioning pulse (*Figure 3F*). The tau

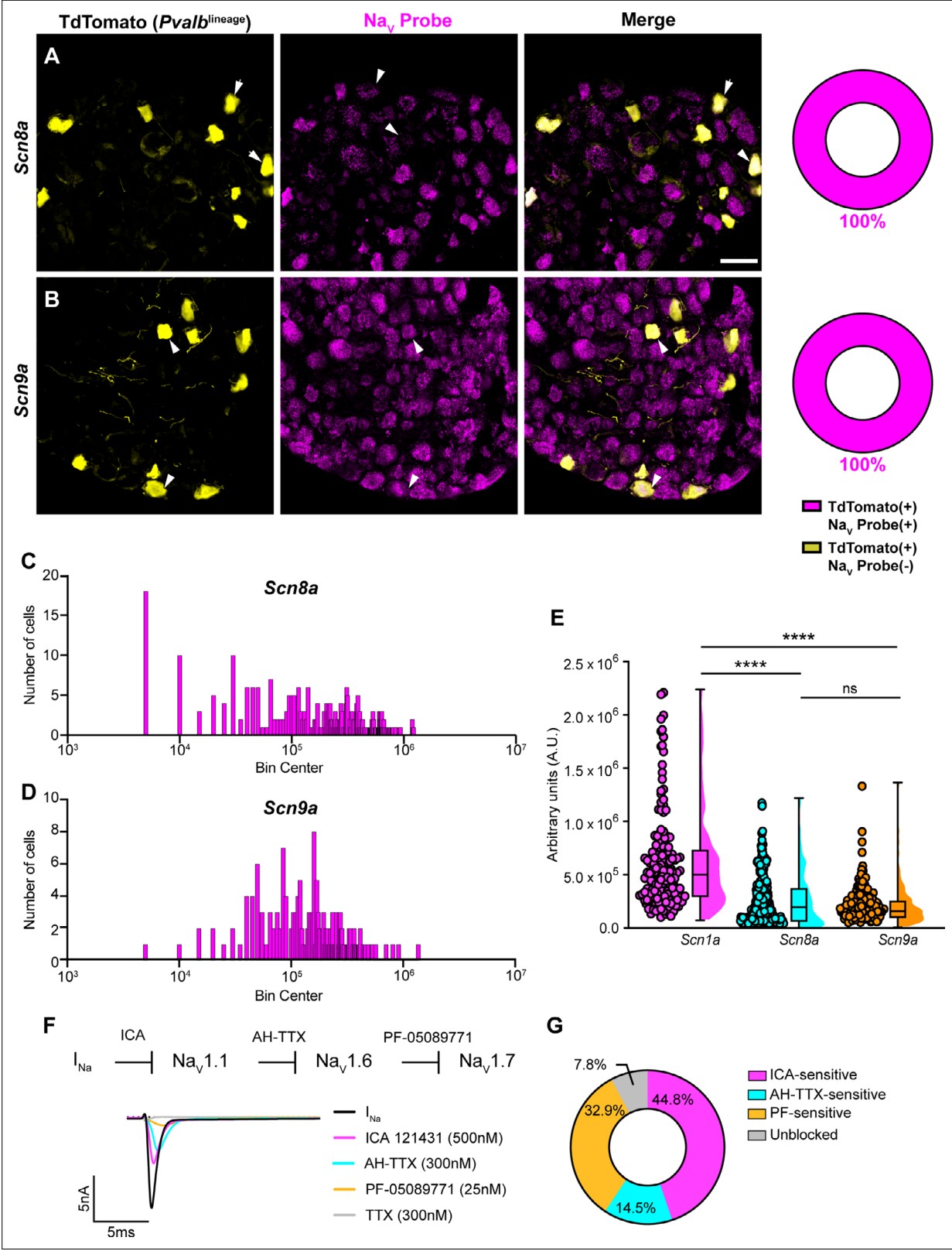

**Figure 2.** Transcriptomic and functional expression of sodium channels in proprioceptors. (**A, B**) Representative confocal images of cryoprotected adult dorsal root ganglia (DRG) sections (25 µm) with quantifications indicating the percentage of Na$_V$+ and Na$_V$- neurons in TdTomato+ proprioceptors. Sections were hybridized using RNAscope with probes targeting Na$_V$1.6 or Na$_V$1.7 (*Scn8a* and *Scn9a*, respectively, magenta) and stained with anti-DsRed (yellow). (**A**) *Snc8a*, n = 298, (**B**) *Scn9a*, n = 166. Scale bar set to 50 µm. White arrowheads indicate Na$_V$+ neurons. (**C–, D**) Frequency distribution

*Figure 2 continued on next page*

*Figure 2 continued*

plots of integrated fluorescence density of Na$_V$1.6 and Na$_V$1.7 mRNA in TdTomato+ proprioceptors, respectively. (**E**) The average integrated density of *Scn1a*, *Scn8a*, and *Scn9a* RNAscope probe signal. Dots represent individual cells. Statistical significance was determined using a Kruskal–Wallis test with Dunn's post-hoc comparisons. (**F**) Top: experimental workflow of serial pharmacological blockade of Na$_V$ channels expressed in proprioceptors. We first elicited a whole-cell sodium current in the absence of drug. We next bath applied 500 nM of ICA 121431 to block current carried by Na$_V$1.1. Subsequently, we bath applied 9-anhydroustetrodoxin (AH-TTX) (300 nM) to block Na$_V$1.6-mediated current, and PF-05089771 (25 nM) to block the Na$_V$1.7-mediated current. Finally, tetrodotoxin (TTX) (300 nM) was used to block residual current and to confirm there was no contribution of TTX-resistant Na$_V$s in proprioceptors. Bottom: representative current traces following application of Na$_V$-selective inhibitors. All drugs were applied for 1 min. (**G**) Quantification of the average percentage of the whole-cell sodium current that was sensitive to the individual drugs used (n = 8). n = cells. ****p<0.0001.

The online version of this article includes the following figure supplement(s) for figure 2:

**Figure supplement 1.** 0.1% DMSO vehicle does not change I$_{Na}$ in proprioceptors.

for entry into slow inactivation was 928.6 ms, with more than 50% of channels available after a 2000 ms conditioning pulse.

We next asked how blocking Na$_V$1.1 channels affects proprioceptor function in vitro. Similar to serial pharmacological experiments, I$_{Na}$ density in proprioceptors was significantly reduced from

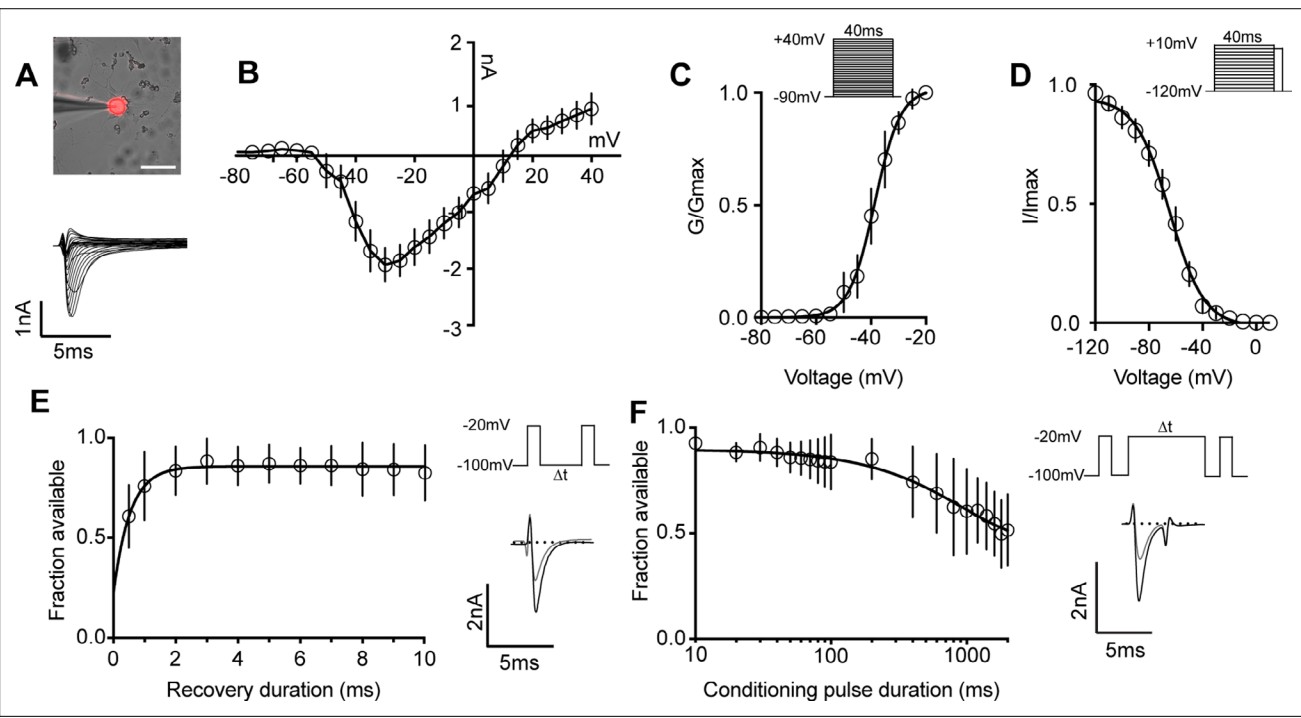

**Figure 3.** Biophysical analysis of the whole-cell sodium current (I$_{Na}$) in genetically identified proprioceptors. (**A**) Top: representative image of a TdTomato+ proprioceptor in culture during electrophysiological recordings (scale bar set to 50 µm). Bottom: representative current traces from current–voltage relationship of I$_{Na}$ from TdTomato+ proprioceptors shown in (**B**). Currents were elicited by 40 ms voltage steps from –90 mV to +40 mV in 5 mV increments (n = 7–9). (**C**) Top: the voltage protocol used to measure the voltage dependence of whole-cell sodium current activation in proprioceptors. Currents were elicited using a series of 40 ms voltage steps from –90 mV to 40 mV at 5 mV increments from a holding potential of –90 mV. Bottom: data are expressed as conductance over maximum conductance (n = 6–9). (**D**) Top: the voltage protocol used to measure the voltage dependence of inactivation. A 40 ms prepulse ranging from –120 mV to +10 mV was given followed by a test pulse to 0 mV. Bottom: data are expressed as current over maximum current (n = 8–12). (**E**) Left: quantification of recovery from fast inactivation (n = 8). Line shows a monoexponential fit of the data (τ = 0.54 ms). Top right: voltage protocol to measure recovery from fast inactivation. A 20 ms step to –20 mV from –100 mV is followed by varying durations of recovery at –100 mV before a second test step to –20 mV. Bottom right: representative traces of currents elicited before (black trace) and after (gray trace) a 0.5 ms recovery period. (**F**) Left: quantification of entry into slow inactivation (n = 6). Line shows a monoexponential fit of the data (τ = 928.6 ms). Top right: voltage protocol to measure entry into slow inactivation. A 3 ms test pulse to –20 mV from –100 mV was followed by conditioning pulses at 0 mV for varying durations before a third test step to –20 mV. 12 ms recovery periods after the first test pulse and before the second were included to remove fast inactivation. Bottom right: representative current trace elicited before and after a 2000 ms conditioning pulse. All error bars represent standarad error of the mean (SEM) n = cells.

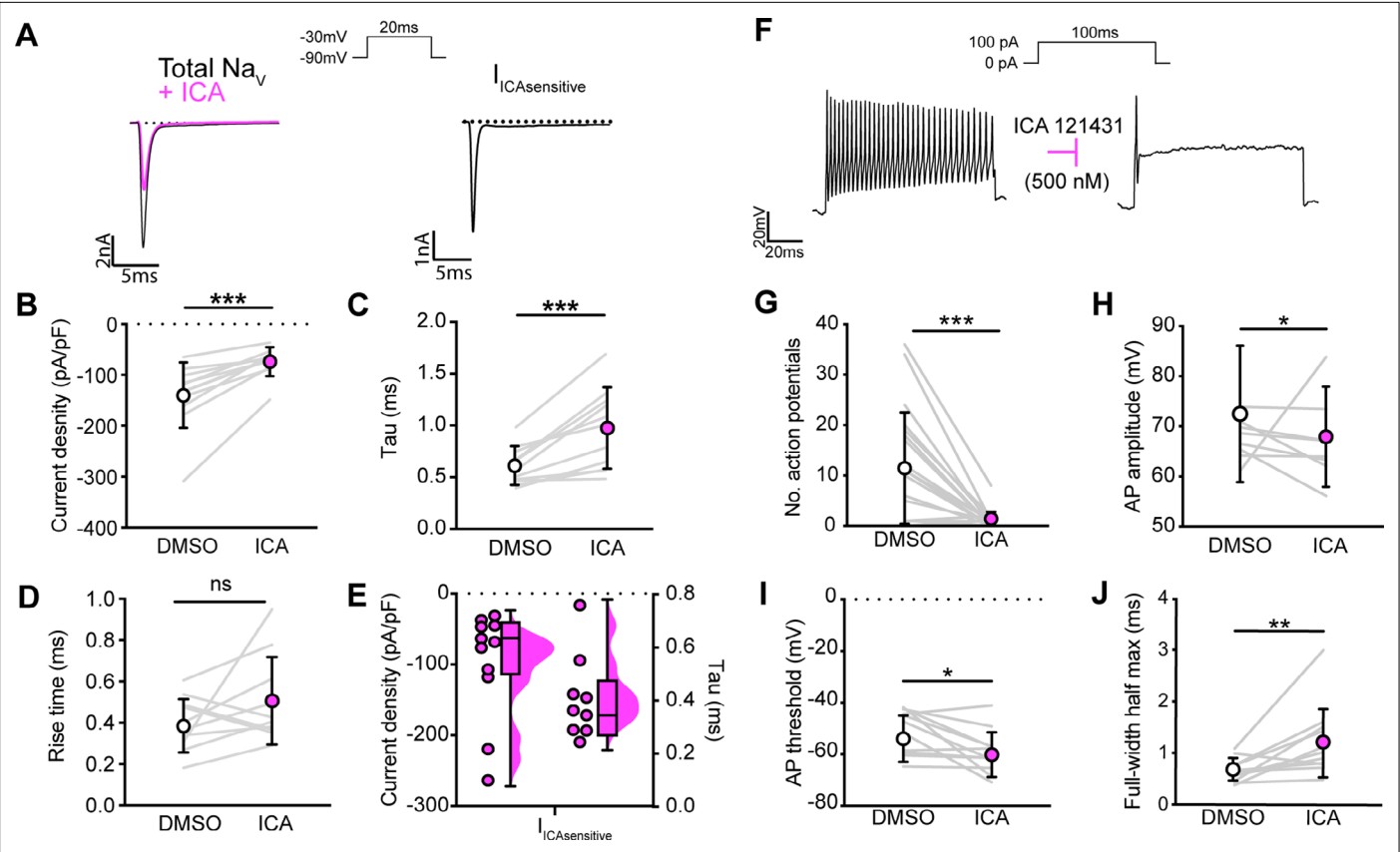

**Figure 4.** Na$_V$1.1 significantly contributes to the whole-cell sodium current and intrinsic excitability in genetically identified proprioceptors. (**A**) Left: representative whole-cell voltage-clamp traces elicited before (black) and after (magenta) application of ICA 121431 (500 nM). Right: the subtracted ICA-sensitive current shown in black. (**B**) Quantification of the reduction in whole-cell current density before (white) and after (magenta) ICA, p=0.0003, n = 11 cells. (**C**) Quantification of rate of current decay before and after ICA, p=0.0007, n = 11 cells. (**D**) Quantification of whole-cell current rise time before and after ICA, p=0.1611, n = 10 cells. (**E**) Left: current densities of ICA-sensitive sodium currents (n = 11); right: current decay taus of ICA-sensitive sodium currents (n = 9 cells). (**F**) Representative whole-cell current-clamp traces before (left) and after (right) application of ICA. (**G–J**) Quantification of number of action potentials in response to current injection (**G**; p=0.0002, n = 20 cells), action potential amplitude (**H**; p=0.0420, n = 20 cells), action potential threshold (**I**; p=0.0186, n = 20 cells), and full-width half max (**J**; p=0.0068, n = 20 cells). Gray lines represent paired observations, circles and lines represent means and standard deviations. White circles, before ICA application. Magenta circles, after ICA application. The Wilcoxon matched-pairs signed-rank test was used to determine statistical significance.

The online version of this article includes the following figure supplement(s) for figure 4:

**Figure supplement 1.** ICA 121431 may inhibit upregulated Na$_V$1.3 channels in cultured dorsal root ganglia (DRG) neurons.

~–140 pA/pF to ~–75 pA/pF following ICA application (*Figure 4B*, p=0.0003). The proprioceptor I$_{Na}$ had an average tau of 0.6 ms, which was significantly slowed to 1.0 ms following application of ICA (*Figure 4C*, p=0.0007), in line with loss of a fast gating channel. Blocking Na$_V$1.1 did not change I$_{Na}$ rise time (*Figure 4D*, p=0.1611). Quantification of the ICA-sensitive component found an average tau of 0.4 ms and an average current density of –96 pA/pF (*Figure 4E*). Of note, there was a wide distribution of current densities for the ICA-sensitive component, ranging from ~–28 pA/pF to ~–263 pA/pF, suggesting some variability in the contribution of Na$_V$1.1 to proprioceptor excitability that may be proprioceptor subtype dependent. We next used current-clamp experiments to determine the effect of ICA on proprioceptor intrinsic excitability (*Figure 4F*). Pharmacological inhibition of Na$_V$1.1 significantly reduced the number of evoked action potentials in most genetically identified proprioceptors (*Figure 4G*); however, five of the cells recorded had low firing rates that were not inhibited by ICA. This further suggests that Na$_V$1.1 is important for repetitive firing in most proprioceptors, but some subtypes with lower intrinsic excitability instead rely on a combination of Na$_V$1.6 and Na$_V$1.7. Action potential amplitude (*Figure 4H*, p=0.0420) and action potential threshold (*Figure 4I*, p=0.0186) were also significantly reduced following ICA application, and action potential full-width half-max was

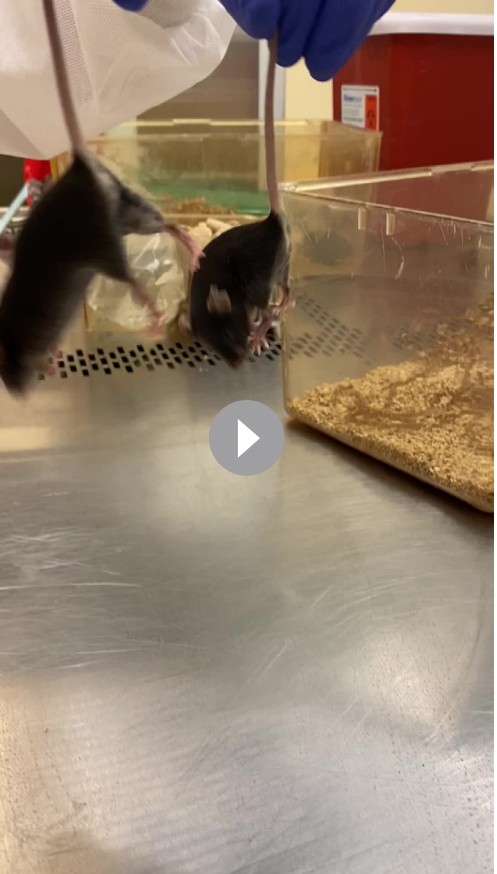

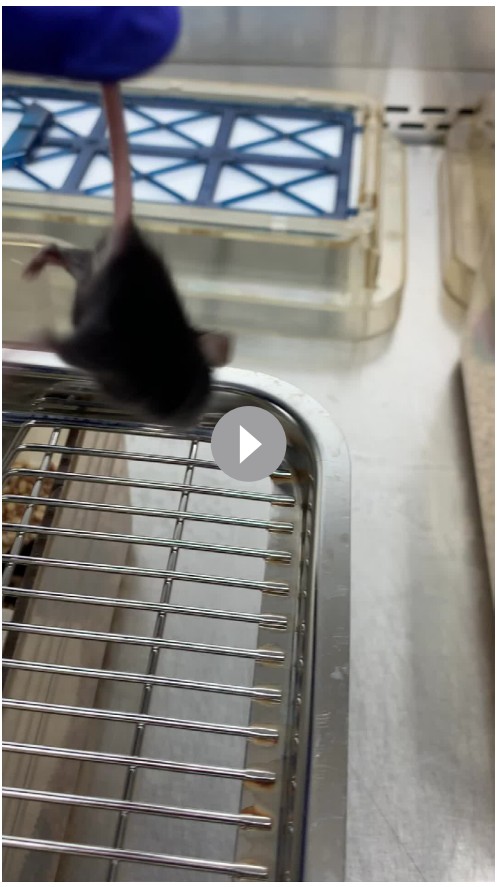

**Video 1.** Uncoordinated movements in Scn1a-cKO animals. A Scn1a-cKO mouse (left) shows abnormal and spastic movements when suspended in the air. These movements are absent in Scn1a-floxed mice (right).
https://elifesciences.org/articles/79917/figures#video1

**Video 2.** Abnormal limb position in Scn1a-cKO animals. A Scn1a-cKO mouse has uncoordinated leg movements and makes an abnormal rotation of its hind paw to grasp its tail while suspended in the air.
https://elifesciences.org/articles/79917/figures#video2

significantly increased following ICA application (*Figure 4K*, p=0.0068), in line with loss of the fast $Na_V1.1$-mediated current.

It is important to note that ICA 121431 also blocks $Na_V1.3$ channels, which could be upregulated in our cultured DRG neuron preparations (*Wangzhou et al., 2020*). Indeed, a small but significant decrease in $I_{Na}$ was observed in recordings from large-diameter DRG neurons harvested from *Pirt*^Cre;Scn1a-floxed mice (Scn1a-cKO), which lack $Na_V1.1$ in all sensory neurons. Thus, inhibition of upregulated $Na_V1.3$ channels could contribute to the effect of on ICA on the proprioceptor $I_{Na}$(-*Figure 4—figure supplement 1*).

To clarify the importance of $Na_V1.1$ to proprioceptor function and avoid the caveats associated with in vitro pharmacological studies, we took an in vivo approach and analyzed motor behaviors in Scn1a-cKO mice of both sexes. We were precluded from using a *Pvalb*^cre driver line to directly interrogate a role for $Na_V1.1$ in proprioceptors as loss of $Na_V1.1$ in Pvalb-expressing brain interneurons produces an epilepsy phenotype that prevents behavioral analyses in adult animals (*Ogiwara et al., 2007*). Consistent with in vitro data, Scn1a-cKO animals of both sexes displayed profound and visible motor abnormalities. These abnormalities include ataxic-like tremors when suspended in the air (*Video 1*), abnormal limb positioning (*Videos 2 and 3*), and paw clasping, which are absent in Scn1a-floxed littermate controls and heterozygous animals (*Pirt*^Cre;*Scn1a*^fl/+, Scn1a-Het, respectively, *Figure 5A*). We first ran animals in the open-field test for 10 min each to quantify spontaneous locomotor behaviors (*Figure 5B*). We found that Scn1a-cKO animals traveled significantly less (*Figure 5C*) and slower (*Figure 5D*) than Scn1a-floxed littermate controls (p=0.0077 and p=0.0057, respectively). Surprisingly, Scn1a-Het mice also displayed motor abnormalities in the open-field test, performing

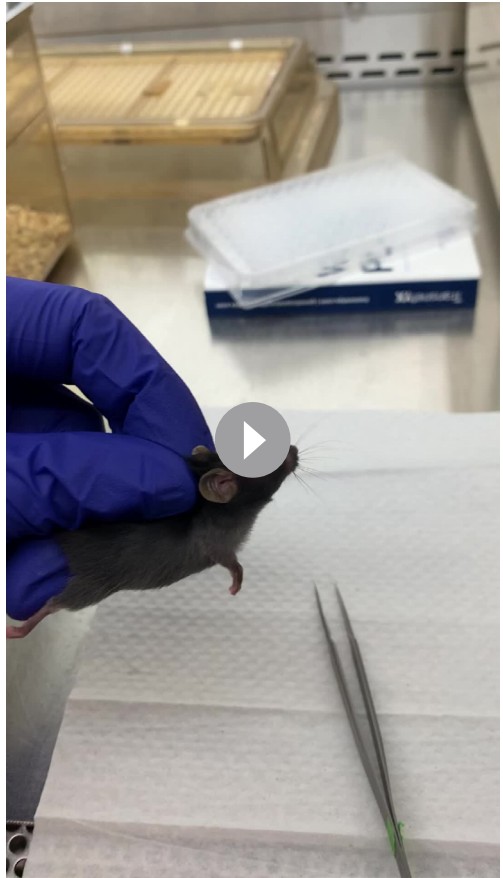

**Video 3.** Abnormal paw position in Scn1a-cKO animals. A Scn1a-cKO mouse is scruffed and places hind paws with foot pads facing down. This contrasts with the normal paw positioning seen in the forepaws, in which foot pads are in the outward-facing position.
https://elifesciences.org/articles/79917/figures#video3

similarly to Scn1a-cKO animals (*Figure 5B–D*), demonstrating $Na_V1.1$ haploinsufficiency in sensory neurons for motor behaviors. No genotype-dependent differences were observed in the amount of time spent moving, suggesting gross motor function was intact (*Figure 5E*). Additionally, the amount of time spent in the center of the open-field chamber was also independent of genotype (*Figure 5F*). We next used the rotarod assay to investigate differences in motor coordination. Mice were assayed on three consecutive days and latency to fall and revolutions per minute (RPM) were quantified. Unlike in the open-field assay, both Scn1a-floxed and Scn1a-Het mice performed at similar levels during the 3-day period (*Figure 5G and H*). Conversely, Scn1a-cKO animals performed significantly worse. By day 3, on average they were only able to maintain their position on the rotarod for 41 s, falling over 50% faster Scn1a-floxed and Scn1a-Het mice. We did not observe any sex-dependent differences in performance in the open-field or rotarod tests (*Figure 5—figure supplement 1*). We confirmed that our mouse model selectively targeted sensory neurons by crossing a $Pirt^{Cre}$ driver with a fluorescent reporter line ($Pirt^{Cre}$;$Rosa26^{Ai14}$). We observed little-to-no neuronal expression of TdTomato in both dorsal and ventral spinal cord (*Figure 5—figure supplement 2*). In contrast, DRG somata and axons showed strong labeling. Collectively, our behavioral data provide evidence for a new in vivo role of $Na_V1.1$ in sensory neurons in mammalian proprioception.

Could the motor deficits observed in Scn1a-cKO mice be due to abnormal proprioceptor development? To address this question, we performed RNAscope analysis of DRG sections from Scn1a-floxed, Scn1a-Het, and Scn1a-cKO mice. We quantified the number of neurons per DRG section that were positive for both *Runx3* and *Pvalb* transcript, the molecular signature of mature proprioceptors (*Figure 6*, *Oliver et al., 2021*). We found no significant genotype-dependent differences in the number of proprioceptors in Scn1a-Het and Scn1a-cKO mice compared to Scn1a-floxed controls (p=0.3824 and p=0.1665, respectively), indicating that the behavioral deficits observed in Scn1a-cKO mice are not the result of a developmental loss of proprioceptors. We also analyzed muscle spindle morphology to determine whether aberrant sensory end organ development may contribute to the observed motor abnormalities. Similar to conditional Piezo2-knockout animals (*Woo et al., 2015*), no qualitative differences were observed between genotypes (*Figure 6—figure supplement 1*). Thus, abnormal proprioceptor development does not contribute to the overall phenotype of Scn1a-cKO mice.

We next asked whether the motor deficits of Scn1a-cKO mice are due to altered synaptic connectivity between proprioceptive axons and motor neurons in the ventral spinal cord. Spinal cord sections were harvested from Scn1a-floxed, Scn1a-Het, and Scn1a-cKO mice and stained with antibodies against vesicular glutamate transporter 1 (VGLUT1) to label proprioceptor axons, and choline acetyltransferase (ChAT). ChAT primarily labels α- and γ-motor neurons in the ventral horn, which can be distinguished based on size. We analyzed the number of VGLUT1 puncta on the somata and proximal dendrites of individual cholinergic neurons greater than 400 μm² to bias our quantification towards

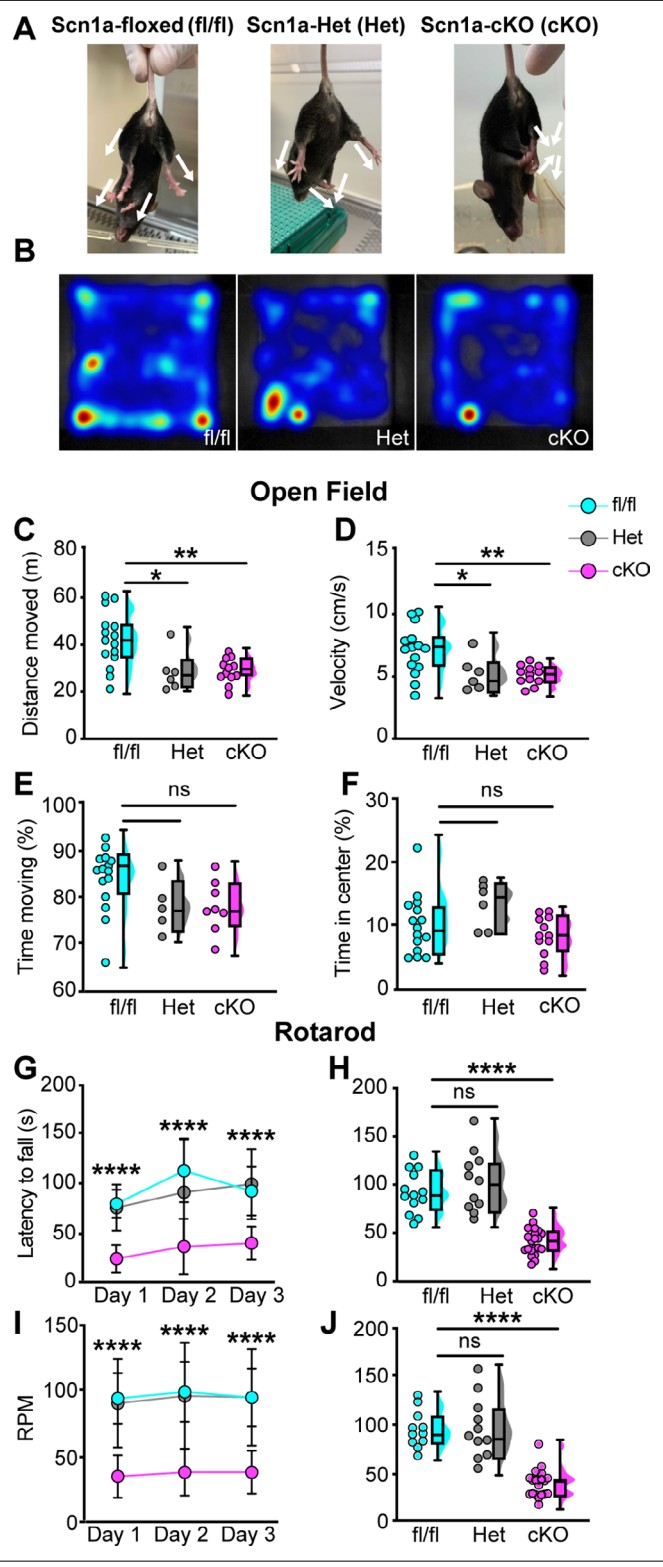

**Figure 5.** Loss of Na$_V$1.1 in peripheral sensory causes deficits in motor behaviors. (**A**) Representative images showing limb positions of adult Scn1a-floxed (left), Scn1a-Het (middle), and Scn1a-cKO (right) mice. White arrows represent the direction of limbs. (**B**) Representative heat maps from open-field experiments between Scn1a-floxed (left), Scn1a-Het (middle), and Scn1a-cKO (right). Open field (**C–F**). Quantification of total distance traveled during a 10-min open-field test between Scn1a-floxed (cyan), Scn1a-Het (gray), and Scn1a-cKO (magenta) mice.

*Figure 5 continued on next page*

*Figure 5 continued*

(**C**) Scn1a-Het p=0.0255, Scn1a-cKO, p=0.0077, compared to Scn1a-floxed, average animal velocity. (**D**) Scn1a-Het p=0.00311, Scn1a cKO, p=0.0057, compared to Scn1a-floxed, percent time moving. (**E**) Scn1a-Het p=0.1362, p=0.0730, compared to Scn1a-floxed, and percent time spent in center. (**F**) Scn1a-Het p=0.2297, Scn1a-cKO, p=0.2494, compared to Scn1a-floxed, during the test. Rotarod (**G–I**). Quantification of the latency to fall across three consecutive training days (**G**) and day 3 (**H**). Quantification of revolutions per minute (RPM) at the moment of animal fall (**I**) and the day 3 average (**J**). Each dot represents one animal. Box and whisker plots represent maximum, minimum, median, upper and lower quartiles of data sets.****p<0.0001. A one-way ANOVA (Dunnett's post-hoc comparison) was used to determine statistical significance in (**C–F**) and (**I**) and (**J**). A two-way mixed-design ANOVA (Dunnett's post-hoc comparison) was used to determine statistical significance in (**G**) and (**H**) Open field: Scn1a-floxed N = 15, Scn1a-Het N = 6, Scn1a-cKO N = 12. Rotarod: Scn1a-floxed N = 11, Scn1a-Het N = 11, Scn1a-cKO N = 20.

The online version of this article includes the following figure supplement(s) for figure 5:

**Figure supplement 1.** Motor deficits in Scn1a-Het and Scn1a-cKO animals are not sex dependent.

**Figure supplement 2.** TdTomato expression is limited to sensory neurons.

α-motorneurons. We found no significant decrease in VGLUT puncta density per ChAT+ neuron between Scn1a-Het or Scn1a-cKO when compared to Scn1a-floxed littermate controls (*Figure 7*, Scn1a-Het, p>0.9999, Scn1a-cKO, p=0.4573). This suggests that general deficits in proprioceptor innervation of motor neurons do not contribute to the phenotype of Scn1a-cKO mice.

We next asked whether proprioceptor electrical signaling is altered in Scn1a-cKO mice. While in vitro patch-clamp electrophysiology can assess $Na_V$ function at DRG somata and provide insight as to how they contribute to intrinsic excitability, the physiological contributions of ion channels in DRG soma to somatosensory transmission in vivo are not well understood. Thus, to directly investigate how $Na_V1.1$ shapes action potential propagation down proprioceptor axons, we used an ex vivo preparation to record muscle afferent activity during ramp-and-hold stretch and sinusoidal vibration. Afferents from both Scn1a-Het and Scn1a-cKO mice exhibited impaired static stretch sensitivity as evidenced by a decreased likelihood of firing during rest as compared to Scn1a-floxed mice, as well as an inability to maintain firing throughout the entire 4 s stretch. Almost all afferents from Scn1a-floxed mice could fire consistently throughout the entire 4 s hold phase (*Figure 8A*), but loss of one or both copies of $Na_V1.1$ led to either firing only near the beginning of stretch or inconsistent firing in a high percentage of afferents lacking $Na_V1.1$ (*Figure 8B and C*). We quantified this inconsistent firing by determining the CV of the interspike interval (ISI) during the plateau phase of stretch (1.5–3.5 s into the hold phase) across different stretch lengths and found a significant effect of genotype, with the knockout afferents both having higher ISI CV than the Scn1a-floxed afferents (*Figure 8D*; 0.074 ± 0.06, 0.313 ± 0.456, 0.497 ± 0.831, at 7.5% Lo, Scn1a-floxed, Scn1a-Het, and Scn1a-cKO afferents, respectively, two-way ANOVA, main effect of genotype, p=0.015). In contrast to the clear deficits in static sensitivity in afferents lacking $Na_V1.1$, dynamic sensitivity was not significantly impaired. The maximum firing frequency during the rampup phase (Dynamic Peak) was independent of genotype, and even trended slightly higher in afferents lacking $Na_V1.1$ (*Figure 8E*; two-way ANOVA, effect of genotype p=0.0633).

We next examined the requirement of $Na_V1.1$ for proprioceptor afferent responses to sinusoidal vibration, which is a measure of dynamic sensitivity, and found no differences with loss of $Na_V1.1$ (*Figure 9A–C*, *Tables 1–3*). We characterized a unit as having entrained to vibration if it fired at approximately the same time every cycle of the 9 s vibration. In most cases, afferents lacking $Na_V1.1$ were equally likely to entrain to vibration than Scn1a-floxed afferents (*Figure 9D–F*). Indeed, Scn1a-cKO afferents were able to maintain firing during the entire 9 s sinusoidal vibration, in contrast to their inability to maintain consistent firing during 4 s of static stretch. There were no significant differences in firing rate during vibration between Scn1a-floxed, Scn1a-Het, and Scn1a-cKO afferents (*Figure 9D–F*). Taken together, our ex vivo recordings suggest that behavioral deficits in Scn1a-cKO result from abnormal proprioceptor responses to static muscle movement, whereas afferent responsiveness to dynamic stimuli is $Na_V1.1$-independent. Furthermore, recordings from Scn1a-Het animals support the notion that $Na_V1.1$ is haploinsufficient for proprioceptor function at the cellular level.

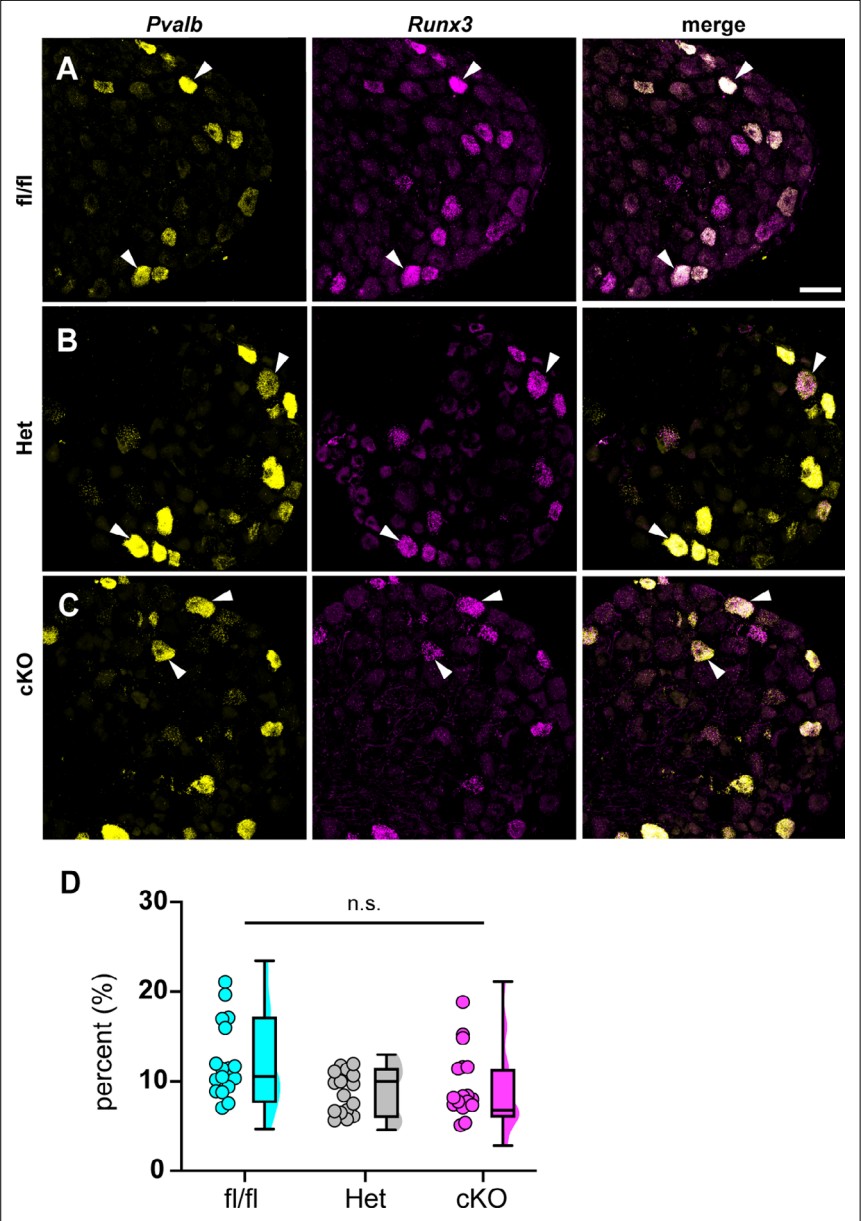

**Figure 6.** Loss of Na$_V$1.1 in sensory neurons does not affect proprioceptor development. Representative images of Scn1a-floxed (**A**), Scn1a-Het (**B**), and Scn1a-cKO (**C**) adult dorsal root ganglia (DRG) neuron sections (25 µm). Images were acquired with a ×40, 0.9 NA water-immersion objective. Sections were hybridized with probes targeting parvalbumin (*Pvalb*, yellow) and *Runx3* (magenta). (**D**) Quantification of the percentage of *Pvalb+*/*Runx3+* neurons in each genotype. Each dot represents one DRG section. Box and whisker plots represent maximum, minimum, median, upper and lower quartiles of data sets. A Kruskal–Wallis test with Dunn's post-hoc comparison was used to determine statistical significance (p=0.1971, Scn1a-floxed n = 17, Scn1a-Het n = 17, Scn1a-cKO n = 18). n = sections. Scale bar 50 µm.

The online version of this article includes the following figure supplement(s) for figure 6:

**Figure supplement 1.** Muscle spindle development is normal in Scn1a-Het and Scn1a-cKO animals.

## Discussion

The critical role for Na$_V$1.1 in various brain disorders has overshadowed the potential contributions of this channel in peripheral signaling. The results presented in this study are the first to provide functional evidence that Na$_V$1.1 in peripheral sensory neurons is required for normal proprioception. We found that mice lacking Na$_V$1.1 in sensory neurons exhibit visible motor deficits and ataxic-like behaviors,

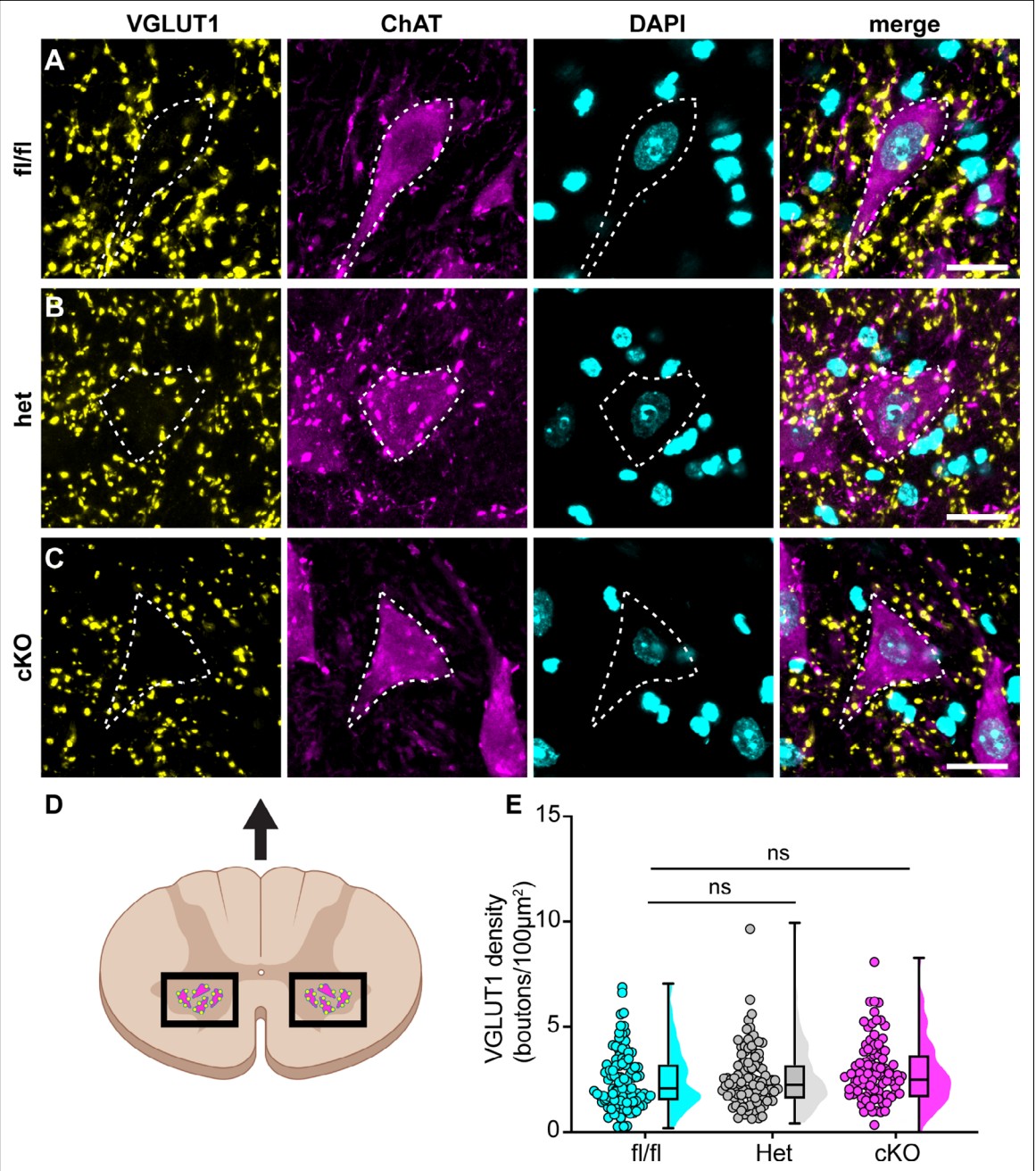

**Figure 7.** Loss of Na$_V$1.1 in sensory neurons does not change proprioceptor innervation of α-motor neurons. (**A–C**) Representative images of Scn1a-floxed (**A**), Scn1a-Het (**B**), and Scn1a-cKO (**C**) adult spinal cord sections (30 μm). Images were acquired with a ×63, 1.4 NA oil-immersion objective. Sections were stained using immunochemistry with VGLUT1 (yellow) and ChAT (magenta). Nuclei (cyan) were labeled with DAPI. (**D**) Schematic of spinal cord regions of interest. (**E**) Quantification of the average density of VGLUT1+ puncta per 100 μm$^2$ onto ChAT+ neurons that were larger than 400 μm$^2$. A Kruskal–Wallis test with Dunn's post-hoc comparison was used to determine statistical significance. Each dot represents a motor neuron. Box and whisker plots represent maximum, minimum, median, upper and lower quartiles of data sets. Scn1a-floxed, n = 101; Scn1a-Het, n = 102; Scn1a-cKO, n = 92. Scale bar 20 μm. n = cells.

which we propose is largely attributed to loss of Na$_V$1.1 in proprioceptors. Indeed, RNAscope analysis showed expression of Na$_V$1.1 mRNA in 100% of genetically identified proprioceptors. While Na$_V$1.6 and Na$_V$1.7 were also ubiquitously expressed in proprioceptors, Na$_V$1.1 displayed higher expression levels, consistent with a previous RNA-sequencing study (*Zheng et al., 2019*). There are anywhere from 5 to 8 different proprioceptor molecular subclasses (*Oliver et al., 2021*; *Wu et al., 2021*), however, and it is possible that these distinct classes rely on different combinations of these channels

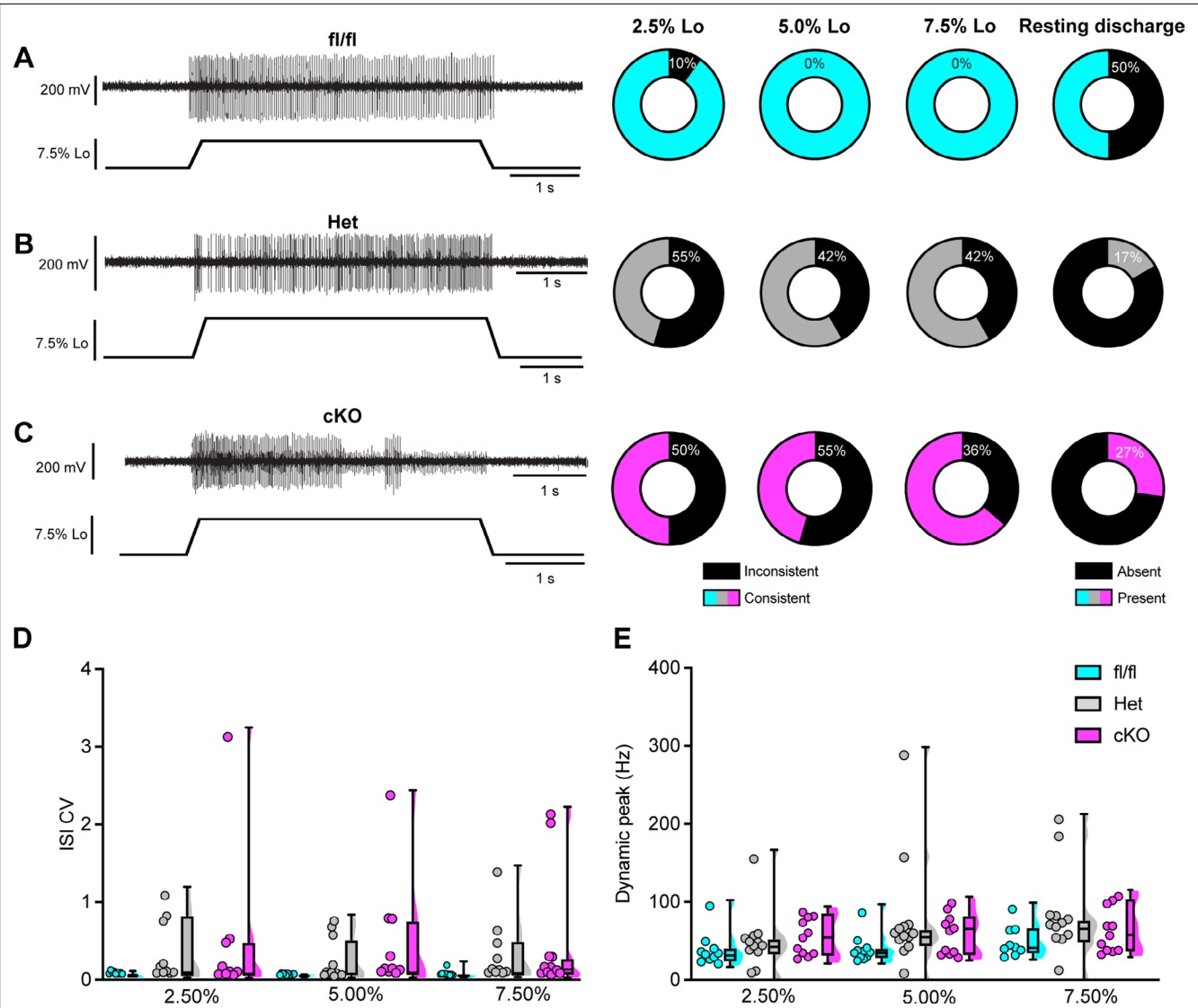

**Figure 8.** Loss of Na$_V$1.1 reduces static muscle stretch sensitivity and reliability. (**A–C**) Representative responses to ramp-and-hold muscle stretch at 7.5% of optimal length (Lo) from Scn1a-floxed (**A**), Scn1a-Het (**B**), and Scn1a-cKO (**C**) afferents. Afferents from Scn1a-Het and Scn1a-cKO mice were more likely to show inconsistent firing during the hold phase of stretch. The percentage of afferents from each genotype that were able to fire consistently for the entire duration of stretch at 2.5, 5.0, and 7.5% of Lo is shown in the pie charts next to the representative trace from their genotype (black indicates percentage with inconsistent firing). The final pie charts represent the proportion of afferents that exhibited resting discharge at Lo for every stretch for each genotype (black indicates absence of resting discharge). (**D**) Inconsistency in firing was quantified as the interspike interval coefficient of variation (ISI CV) during the plateau stage of the hold phase of stretch (1.5–3.5 s into stretch) for the three different genotypes. A significant effect of genotype was observed (two-way mixed-design ANOVA, p=0.015). (**E**) The highest firing rate during the rampup phase of stretch (dynamic peak) is a measure of dynamic sensitivity. No significant effect of genotype on dynamic peak was observed (two-way mixed-design ANOVA, p=0.0633). Box and whisker plots represent maximum, minimum, median, upper and lower quartiles of data sets Each dot represents one afferent in (**D**) and (**E**) (Scn1a-floxed, n = 10; Scn1a-Het, n = 12; Scn1a-cKO, n = 11).

for function. Nevertheless, our functional in vitro patch-clamp experiments found Na$_V$1.1 to be the dominant subtype across recorded neurons, comprising nearly half of the proprioceptor I$_{Na}$. In line with this, pharmacological inhibition of Na$_V$1.1 is sufficient to significantly attenuate action potential firing in most proprioceptors; yet, it should be noted that 25% of the neurons we recorded fired action potentials that were insensitive to ICA application. This suggests that some proprioceptor subtypes rely more heavily on Na$_V$1.6 and Na$_V$1.7 for electrical activity. Interestingly, the proprioceptors that were insensitive to ICA application also were less intrinsic excitable, firing only 1–2 action potentials in response to current injection, as opposed to the other 75% on neurons recorded, which

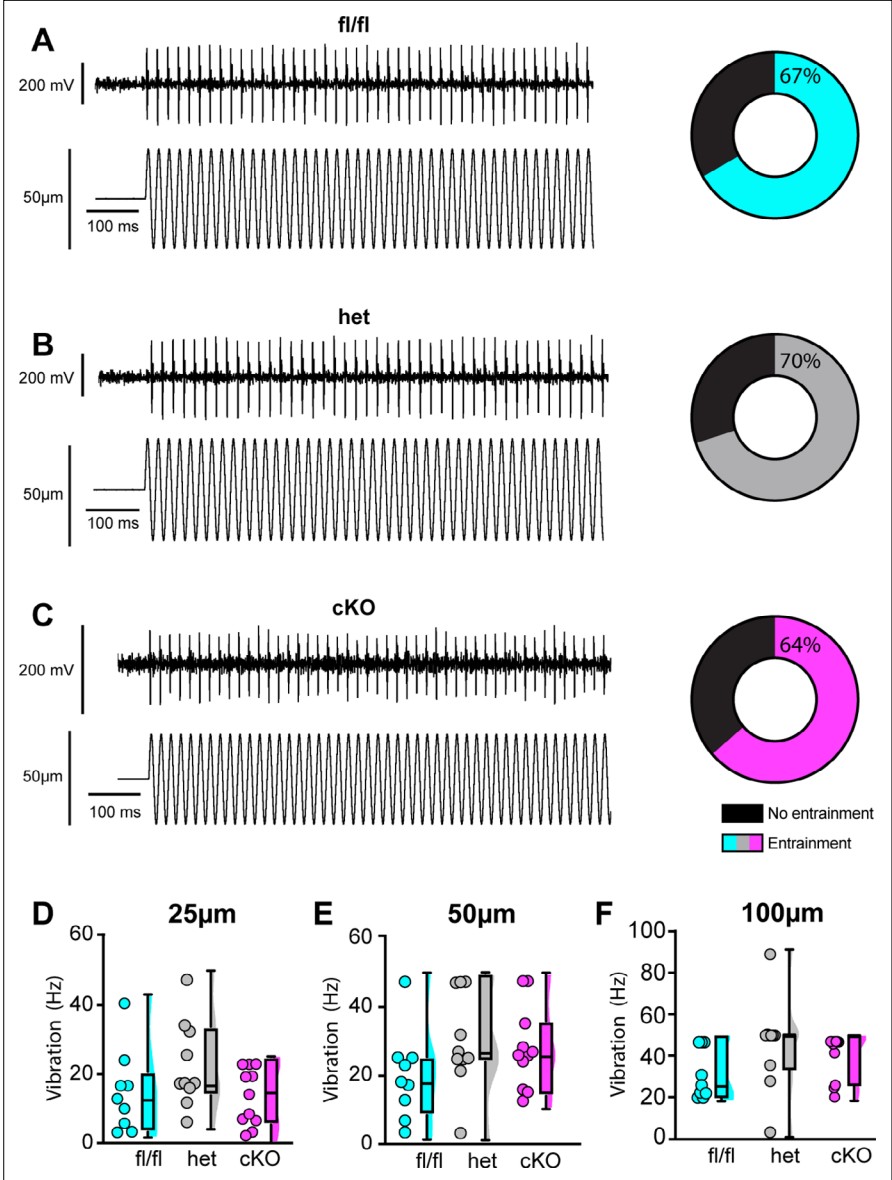

**Figure 9.** Loss of Na$_V$1.1 does not alter muscle spindle afferent response to vibratory muscle stretch. (A–F) Representative traces from afferents that were able to entrain to a 50 Hz, 100 µm vibration as well as graphs with the percentage of all Scn1a-floxed (cyan; A), Scn1a-Het (gray; B), and Scn1a-cKO (magenta; C) afferents that could entrain to the vibration shown in (A–C). Average firing frequency during a 9 s 50 Hz vibration shown for a 25 µm (D), 50 µm (E), and 100 µm (F) amplitude vibration. There was no significant effect of genotype on the firing frequency during vibration (25 µm, p=0.2398, 50 µm, p=0.2413, 100 µm, p=0.1276). A one-way ANOVA was used to determine statistical significance in (D) and (E). A Kruskal–Wallis test was used to determine statistical significance in (F). Box and whisker plots represent maximum, minimum, median, upper and lower quartiles of data sets. Each dot represents one afferent (Scn1a-floxed, n = 9; Scn1a-Het, n = 10; Scn1a-cKO, n = 11).

fired repetitively during the 100 ms injection protocol. This further supports a role for Na$_V$1.1 in maintaining action potential firing in response to sustained stimulation. These results, however, should be interpreted with the caveat that in these experiments ICA may also be acting on Na$_V$1.3 channels that are upregulated during DRG neuron culturing (*Wangzhou et al., 2020*). Nevertheless, at the afferent level Scn1a-cKO and Scn1a-Het animals show clear deficits in static stretch sensitivity, but not dynamic sensitivity, and could even entrain to vibrations as fast as 100 Hz, suggesting a specific role for Na$_V$1.1 in proprioceptors responses to static muscle movement. Finally, we found that loss of Na$_V$1.1 in sensory neurons had no effect on proprioceptor development, muscle spindle morphology,

**Table 1.** Afferent entrainment to 25 μm amplitude vibration.
The percentage of muscle spindle afferents that entrained to a 25 μm amplitude sinusoidal vibration.

| Genotype | 10 **Hz (%)** | 25 **Hz (%)** | 50 **Hz (%)** | 100 **Hz (%)** |
|---|---|---|---|---|
| Scn1a-floxed | 33.33 | 11.11 | 0.00 | 0.00 |
| Scn1a-Het | 50.00 | 40.00 | 10.00 | 10.00 |
| Scn1a-cKO | 27.27 | 9.09 | 0.00 | 0.00 |

or proprioceptive afferent innervation of ChAT+ motor neurons in the spinal cord. Thus, the observed motor behavioral deficits are likely due to reduced static sensitivity of proprioceptor afferents.

Our model proposes that Na$_V$1.1 is tasked with maintaining consistent firing in spindle afferents during static muscle stretch for normal motor behaviors, whereby activation of the mechanotransduction channel Piezo2 initiates electrical signaling, which in turn activates a complement of TTX-sensitive Na$_V$ channels (*Carrasco et al., 2017*; *Florez-Paz et al., 2016*; *Woo et al., 2015*; *Figure 10*). During dynamic or vibratory stimuli, Piezo2, and likely a combination of other molecular mediators, including Na$_V$1.6 and Na$_V$1.7, are sufficient to elicit normal electrical activity. Conversely, during prolonged muscle stretch when Piezo2 channels presumably inactivate, Na$_V$1.1 is required for regular and reliable firing. While other signaling molecules and channels, such as vesicle-released glutamate (*Bewick et al., 2005*; *Than et al., 2021*), and mechanosensitive ASIC channels (*Lin et al., 2016*) and ENaC channels (*Bewick and Banks, 2015*), also contribute to mammalian proprioception, our data suggests that Na$_V$1.1 is a critical for muscle spindle afferent mechanotransduction, given the overt behavioral deficits observed in Scn1a-cKO mice. Due to the ubiquitous expression of Na$_V$1.1 in all proprioceptors and the importance of Golgi tendon organ (GTO) feedback to motor control, alterations in function in those proprioceptors likely contribute to the behavioral deficits we observed; however, we did not directly measure their function.

To date, our knowledge of the functional contributions of Na$_V$1.1 in the PNS is limited. Most studies have identified roles for this channel in mechanical pain signaling in DRG, TG, and vagal sensory neurons. Intraplantar pharmacological activation of Na$_V$1.1 induces spontaneous pain behaviors and mechanical pain, which are absent in mice lacking Na$_V$1.1 in small- and medium-diameter sensory neurons (*Osteen et al., 2016*). Inhibition of Na$_V$1.1 prevented the development of mechanical pain in several preclinical models, including spared nerve injury (*Salvatierra et al., 2018*), an irritable bowel syndrome mouse model (*Salvatierra et al., 2018*), and infraorbital nerve chronic constriction injury (*Pineda-Farias et al., 2021*). Additionally, blocking Na$_V$1.1 channels inhibited firing in TRPM8-expressing neurons in vitro, suggesting a potential role for this channel in thermosensation (*Griffith et al., 2019*). No prior studies, however, have reported a functional role for Na$_V$1.1, or Na$_V$s in general, in proprioception.

The loss of consistent firing we observed during static stretch in Scn1a-cKO and Scn1a-Het animals is functionally similar to deletion of Na$_V$1.1 in other brain cell types. Indeed, loss of a single copy of Na$_V$1.1 is sufficient to attenuate sustained action potential firing in parvalbumin-positive hippocampal interneurons (*Ogiwara et al., 2007*; *Yu et al., 2006*) and cerebellar Purkinje neurons (*Yu et al., 2006*). Na$_V$1.1 has been associated with persistent sodium current (I$_{Na}$P) and resurgent sodium current (I$_{Na}$R), both of which promote repetitive firing in a wide variety of cell types in the CNS and PNS (*Barbosa et al., 2015*; *Kalume et al., 2007*; *Khaliq et al., 2003*). Na$_V$1.1 also recovers rapidly from fast inactivation compared to other channel subtypes (*Herzog et al., 2003*; *Patel et al., 2015*) and has been shown to be refractory to entry into slow inactivation in TRPM8-expressing DRG neurons (*Griffith*

**Table 2.** Afferent entrainment to 50 μm amplitude vibration.
The percentage of muscle spindle afferents that entrained to a 50 μm amplitude sinusoidal vibration.

| Genotype | 10 **Hz (%)** | 25 **Hz (%)** | 50 **Hz (%)** | 100 **Hz (%)** |
|---|---|---|---|---|
| Scn1a-floxed | 88.89 | 22.22 | 11.11 | 11.11 |
| Scn1a-Het | 80.00 | 70.00 | 30.00 | 10.00 |
| Scn1a-cKO | 45.45 | 54.55 | 18.18 | 0.00 |

**Table 3.** Afferent entrainment to 100 µm amplitude vibration.
The percentage of muscle spindle afferents that entrained to a 100 µm amplitude sinusoidal vibration.

| Genotype | 10 **Hz (%)** | 25 **Hz (%)** | 50 **Hz (%)** | 100 **Hz (%)** |
|---|---|---|---|---|
| Scn1a-floxed | 100.00 | 44.44 | 33.33 | 22.22 |
| Scn1a-Het | 90.00 | 90.00 | 70.00 | 40.00 |
| Scn1a-cKO | 63.64 | 72.73 | 63.64 | 18.18 |

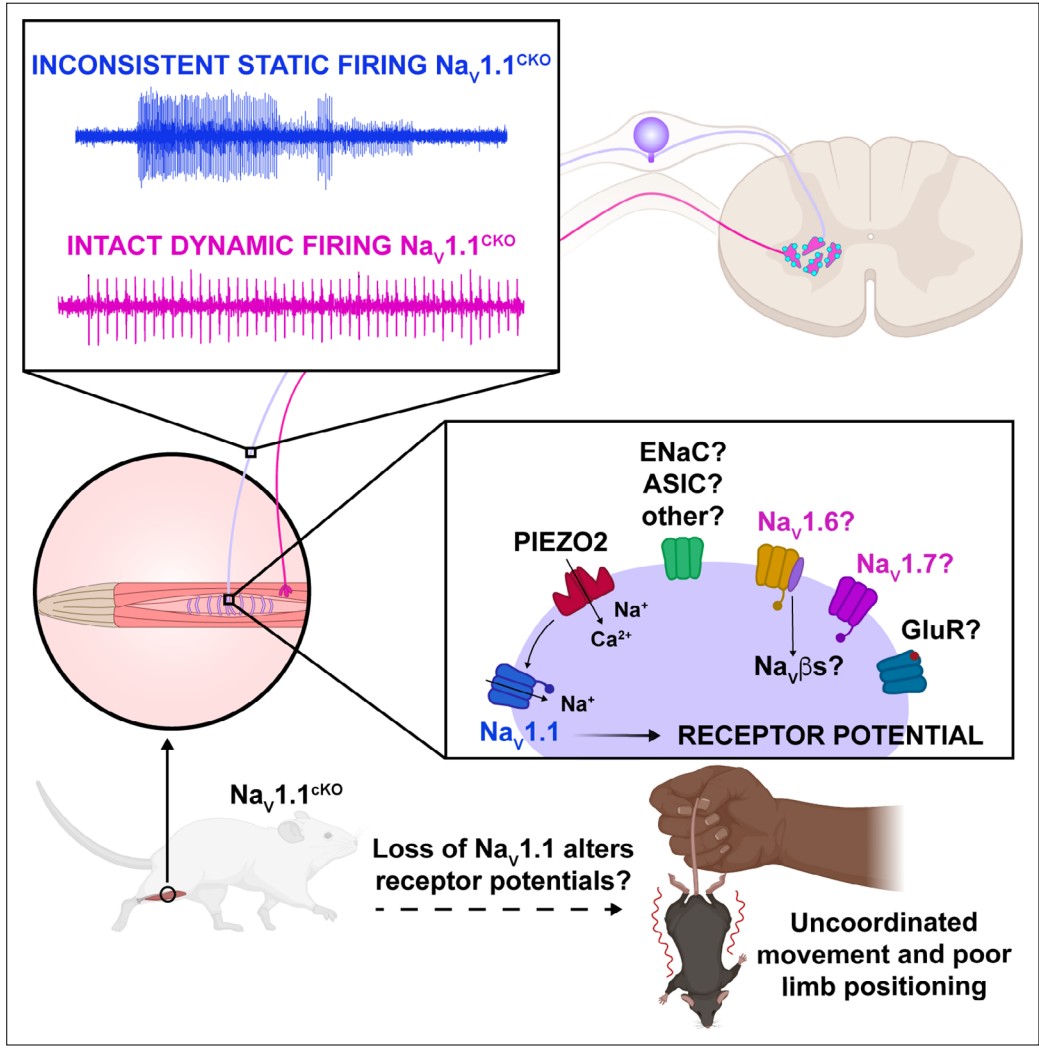

**Figure 10.** Proposed model of the role of Na$_V$1.1 in proprioception. Upon muscle static stretch, various channels activate, including Piezo2 (red), which results in an influx of calcium and sodium ions, causing a depolarization that activates Na$_V$1.1 (dark blue). Na$_V$1.1 activation drives reliable repetitive firing of proprioceptors during static stretch for normal motor behavior. Loss of Na$_V$1.1 in sensory neurons results in inconsistent static firing at the afferent level while maintaining dynamic firing, resulting in uncoordinated movements and abnormal limb positioning. It is possible that a combination of Piezo2, Na$_V$1.6 (yellow), Na$_V$1.7 (pink), and/or other channels, such as glutamate receptors (dark blue), ASIC, and ENaC channels (green), mediates dynamic firing.

*et al., 2019*). We observed similar characteristics when analyzing the proprioceptor $I_{Na}$. Future studies will determine whether proprioceptors rely on these features of $Na_V1.1$ for reliable and consistent encoding of static muscle stretch.

Loss of $Na_V1.1$ notably impacted proprioceptor afferent static sensitivity during ramp-and-hold stretch, but not dynamic sensitivity as measured by entrainment to sinusoidal vibrations using ex vivo muscle-nerve recordings. Afferents from Scn1a-cKO animals were more likely to have action potential failures and thus were largely unable to fire consistently throughout the 4 s of stretch, which was accompanied by a higher coefficient of variability in the ISI. This indicates that $Na_V1.1$ has a critical role in transmitting static stretch information to the CNS. Interestingly, however, dynamic sensitivity in these afferents was unimpaired. Both Scn1a-cKO and Scn1a-Het afferents were able to entrain throughout the entire 9 s vibration; therefore, $Na_V1.1$ does not appear to have a generalized role in maintaining high-frequency firing, but a more specific contribution to muscle-afferent static sensitivity. $Na_V1.1$ has been localized to muscle spindle afferent endings and has been hypothesized to help amplify receptor current (*Carrasco et al., 2017*). Our results support a model whereby current from Piezo2 and potentially other mechanically sensitive ion channels at the start of stretch produces a sufficient receptor potential to generate firing at the heminode, but that amplification of the receptor potential by $Na_V1.1$ is necessary to maintain firing during held stretch.

A similar deficit in static but not dynamic sensitivity was seen following loss of synaptic-like vesicle released glutamate from afferent endings (*Than et al., 2021*); however, in those afferents, firing only occurred at the beginning of stretch and patchy firing was never observed. This may indicate that glutamate plays a more general role maintaining excitability, whereas $Na_V1.1$ is required for reliable action potential generation at heminodes during static stimuli. Alternatively, or in addition, $Na_V1.1$ expressed along the axon could be essential for sustained static firing. A detailed examination of $Na_V1.1$ subcellular localization along proprioceptor afferents could shed light on how this channel contributes to signal propagation. The lack of an effect on dynamic sensitivity could suggest the upregulation of other $Na_V$ subtypes or other molecules as a compensatory mechanism to counteract the loss of $Na_V1.1$. Indeed, our in vitro electrophysiological experiments found a more pronounced effect of acute $Na_V1.1$ inhibition on proprioceptor excitability. This could be due, however, to artificially upregulated $Na_V1.3$ channel activity in culturing conditions, or conversely, a higher density of $Na_V1.1$ expression in proprioceptor somata. Future studies using temporally controlled deletion of $Na_V1.1$ in sensory neurons could tease this apart. Nevertheless, as static sensitivity is still very much impaired in both Scn1a-cKO and Scn1a-Het afferents, $Na_V1.1$ may play a potentially unique role in maintaining afferent firing during the sustained stretch.

Loss of $Na_V1.1$ in sensory neurons did not impact proprioceptor development as the number of proprioceptors in DRG sections was unchanged between genotypes, and muscle spindles developed normally in our model. While we did not directly examine GTOs, we do not anticipate that abnormal end organ development would be restricted to that proprioceptor subtype. Additionally, we did not observe a general decrease in α-motor neuron innervation, which is consistent with the findings of *Mendelsohn et al., 2015*, who reported loss of proprioceptor activity does not generally reduce proprioceptive input into the spinal cord. Interestingly, however, the authors did observe changes in heteronymous sensory-motor connectivity when proprioceptor transmission was blocked, whereby proprioceptor innervation of motor neurons that project to antagonistic muscles was increased. Whether loss of $Na_V1.1$ in proprioceptive afferents causes them to 'mis-wire' in our model to make connections with inappropriate motor neuron pools is unclear. Future studies will determine whether this contributes to the observed behavioral deficits.

We found effects of both pharmacological inhibition and genetic deletion of $Na_V1.1$ in in vitro and ex vivo electrophysiological experiments, respectively (*Figures 2, 4, 8 and 9*); however, in our mouse model $Na_V1.1$ is deleted in all sensory neurons. Thus, we cannot rule out that loss of $Na_V1.1$ in other mechanosenory neuron populations, such as touch receptors, contributes to the motor deficits observed. Deletion of $Na_V1.1$ in small- and medium-diameter DRG neurons using a peripherin-Cre driver did not produce visible motor deficits (*Osteen et al., 2016*), indicating that sensory neuron populations in those categories are not involved. We did observe $Na_V1.1$ transcripts in the vast majority of myelinated DRG neurons (a combination of large- and medium-diameter DRG neurons), consistent with its presence in different subclasses of tactile sensory neurons (*Zheng et al., 2019*). However, the severe motor phenotype of Scn1a-cKO mice precludes mechanical threshold analysis

using von Frey or tactile sensitivity using tape test. Notably, baseline mechanical thresholds were unchanged following intraplantar injection of a selective Na$_V$1.1 inhibitor (*Salvatierra et al., 2018*). This suggests that while Na$_V$1.1 mRNA is expressed in most tactile sensory neurons, functional protein may only be upregulated in these populations during pathological states.

Despite this limitation, one noteworthy and intriguing finding from our study was the haploinsufficiency of Na$_V$1.1 in sensory neurons for proprioceptor function and normal motor behavior in the open-field test. At the afferent level, heterozygous and homozygous loss of Na$_V$1.1 produced similar deficits in static firing, suggesting that loss of less than a quarter of the proprioceptor I$_{Na}$ is sufficient to impair proprioceptor responsiveness to muscle stretch. Na$_V$1.1 is haploinsufficient in several brain neuron cell types for normal excitability and function (*Ogiwara et al., 2007*; *Yu et al., 2006*), suggesting the contributions of Na$_V$1.1 to neuronal function are highly sensitive to genetic perturbations. At the behavioral level, Scn1a-Het mice had an identical phenotype to Scn1a-cKO mice, moving more slowly and less than controls, despite not having the more severe and visible motor coordination deficits. Indeed, their performance on the rotarod was identical to that of Scn1a-floxed controls (*Figure 5*). How these behavioral differences arise given the similar transmission deficits in Scn1a-cKO and Scn1a-Het afferents is unclear. One possibility is a presynaptic role for Na$_V$1.1 in proprioceptive terminals that is unveiled when both copies of Na$_V$1.1 are lost. For example, loss of presynaptic Na$_V$1.7 channels in the spinal cord reduced glutamate release from nociceptive afferents onto dorsal horn neurons (*MacDonald et al., 2021*). If a similar mechanism is at play for Na$_V$1.1 in proprioceptors, reduced neurotransmitter release from Scn1a-Het afferent terminals could be sufficient to produce quantifiable, albeit more subtle, motor deficits. Future studies are required to test this possibility.

Notably, *Scn1a* is a super culprit gene with over 1000 associated disease-causing mutations, most of which are linked to different forms of epilepsy. Many epilepsy patients with hemizygous Na$_V$1.1 loss of function display ataxia and motor delays and deficiencies (*Claes et al., 2001*; *Fujiwara et al., 2003*), which has traditionally been attributed to loss of Na$_V$1.1 function in the brain, namely, the cerebellum (*Kalume et al., 2007*). Our findings suggest that some of the clinical manifestations associated with epilepsy are not solely due to Na$_V$1.1 loss of function in the brain, but also may manifest in part as a result of unreliable coding by peripheral proprioceptors.

Data presented in this study provide new evidence of a role for peripherally expressed Na$_V$1.1 in motor coordination. We show that Na$_V$1.1 is ubiquitously and strongly expressed in proprioceptors, contributes to proprioceptor excitability in vitro and ex vivo, and is haploinsufficient in sensory neurons for normal motor behaviors. Collectively, this work identifies a new role for Na$_V$1.1 in mammalian proprioception.

## Materials and methods

Key Resources Table contains a list of key resources and supplies used for this study.

### Key resources table

| Reagent type (species) or resource | Designation | Source or reference | Identifiers | Additional information |
|---|---|---|---|---|
| Antibody | Anti-DsRed (rabbit polyclonal) | Takara Bio | Cat #632496 | 1:3000 |
| Antibody | GFP (chicken polyclonal) | Abcam | Cat #ab13970 | 1:3000 |
| Antibody | NFH (chicken polyclonal) | Abcam | Cat #ab4680 | In muscle spindles: (1:300) In DRG: (1:3000) |
| Antibody | CGRP (rabbit polyclonal) | ImmunoStar | Cat #24112 | 1:1000 |
| Antibody | VGLUT1 (guinea pig polyclonal) | Zuckerman institute (Columbia University) | Cat #CU1706; RRID:AB_2665455 | In spinal cord: (1:8000) In muscle spindles: (1:800) |
| Antibody | β3-tubulin (chicken polyclonal) | Abcam | Cat #ab41489 | 1:500 |
| Antibody | β3-tubulin (rabbit polyclonal) | Abcam | Cat #ab18207 | 1:3000 |

*Continued on next page*

*Continued*

| Reagent type (species) or resource | Designation | Source or reference | Identifiers | Additional information |
| --- | --- | --- | --- | --- |
| Antibody | ChAT (rabbit polyclonal) | Zuckerman Institute (Columbia University) | Cat #CU1574 | 1:10,000 |
| Chemical compound, drug | VECTASHIELD Antifade Mounting Media with DAPI | Vector Laboratories | Cat #H-2000 | |
| Chemical compound, drug | Tissue-Tek OCT compound | Sakura | Cat #4583 | |
| Chemical compound, drug | Laminin | Sigma-Aldrich | Cat #L2020-1MG | |
| Chemical compound, drug | Collagenase type P | Sigma-Aldrich | Cat #11213865001 | |
| Chemical compound, drug | TrypLE Express | Thermo Fisher | Cat #12605-010 | |
| Chemical compound, drug | MEM | Thermo Fisher | Cat #11095-080 | |
| Chemical compound, drug | Penicillin-streptomycin | Thermo Fisher | Cat #15140-122 | |
| Chemical compound, drug | MEM vitamin solution | Thermo Fisher | Cat #11120-052 | |
| Chemical compound, drug | B-27 supplement | Thermo Fisher | Cat #17504-044 | |
| Chemical compound, drug | Horse serum, heat inactivated | Thermo Fisher | Cat #26050-070 | |
| Chemical compound, drug | ICA 121431 | Tocris Bioscience | Cat #5066/10 | |
| Chemical compound, drug | 4,9-anhydrous tetrodotoxin | Tocris Bioscience | Cat #6159 | |
| Chemical compound, drug | PF-05089771 | Tocris Bioscience | Cat #5931 | |
| Chemical compound, drug | Tetrodotoxin | Abcam | Cat ab120054 | |
| Commercial assay or kit | RNAscope Fluorescence Multiplex Kit | Advanced Cell Diagnostics | Cat #320851 | |
| Sequence-based reagent | *Pvalb* probe channel 1 | Advanced Cell Diagnostics | Cat #421931 | |
| Sequence-based reagent | *Scn1a* probe channel 2 | Advanced Cell Diagnostics | Cat #556181-C2 | |
| Sequence-based reagent | *Scn8a* probe channel 2 | Advanced Cell Diagnostics | Cat #313341-C2 | |
| Sequence-based reagent | *Scn9a* probe channel 2 | Advanced Cell Diagnostics | Cat #434191-C2 | |
| Sequence-based reagent | *Runx3* probe channel 3 | Advanced Cell Diagnostics | Cat #451271-C3 | |
| Strain, strain background (*Mus musculus*) | *Pirt*[cre] | Dr. Xinzhong Dong | | |
| Strain, strain background (*M. musculus*) | *Rosa26*[Ai14] | Jackson Laboratories | Stock #007914 | |

*Continued on next page*

*Continued*

| Reagent type (species) or resource | Designation | Source or reference | Identifiers | Additional information |
| --- | --- | --- | --- | --- |
| Strain, strain background (*M. musculus*) | *Pvalb*cre | Jackson Laboratories | Stock #008069 | |
| Strain, strain background (*M. musculus*) | Scn1a-floxed | UC Davis MMRRC | Stock # 041829-UCD | |
| Strain, strain background (*M. musculus*) | *Mrgprd*GFP | *Zheng et al., 2019* | | |
| Software, algorithm | pClamp 11.2 Software Suite | Molecular Devices | https://www.moleculardevices.com | |
| Software, algorithm | ImageJ | *Schneider et al., 2012* | https://imagej.nih.gov | |
| Software, algorithm | Prism 9 | GraphPad | https://www.graphpad.com | |
| Software, algorithm | LabChart | ADInstruments | https://www.adinstruments.com/products/labchart | |
| Software, algorithm | MATLAB | MathWorks | https://www.mathworks.com | |
| Software, algorithm | MATLAB | This study | https://github.com/doctheagrif/Current-Clamp-Matlab-Code_O-Neil-DA; *O'Neil, 2022b* | Current-clamp experiments were analyzed with a custom-written MATLAB Script that is available on GitHub |

## Animals

*Pirt*cre mice were a kind gift from Dr. Xinzhong Dong (Johns Hopkins University). *Rosa26*Ai14 (stock #007914; *Madisen et al., 2010*) and *Pvalb*cre (stock # 008069) were obtained from Jackson Laboratories. Scn1a-floxed (stock #041829-UCD) mice were purchased from the UC Davis MMRRC. *Mrgprd* mice were originally published in *Zheng et al., 2019*. All mice used were on a mixed C57BL/6 background (non-congenic). Genotyping was outsourced to Transnetyx. Animal use was conducted according to guidelines from the National Institutes of Health's Guide for the Care and Use of Laboratory Animals and was approved by the Institutional Animal Care and Use Committee of Rutgers University-Newark (PROTO201900161), UC Davis (#21947 and #22438), and San José State University (#990, ex vivo muscle recordings). Mice were maintained on a 12 hr light/dark cycle, and food and water was provided ad libitum.

## Rotarod

To assess motor coordination, a rotarod machine (IITC Life Sciences, Woodland Hills, CA) that has an accelerating rotating cylinder was used. 8–10-week-old mice of both sexes were acclimated to the behavior room for 2 hr prior to testing. Mice were assayed on the rotarod for three consecutive days, with three trials per day and an intertrial interval of at least 15 min. The average of the three trials per day was used. The experimenter was blind to genotype.

## Open-field test

8–10-week-old mice of both sexes were acclimated to the behavior room for 2 hr prior to testing. The open-field apparatus consisted of a black square sound attenuating box of dimensions 40.6 cm × 40.6 cm. A camera suspended above the arena was connected to a computer running Ethovision XT software, which tracked animal movement and velocity. An animal was placed in the center of the arena and allowed to freely explore for a 10-min trial. The experimenter was blind to genotype.

## Tissue processing

For spinal cord immunolabeling experiments, whole spinal columns from adult (8–15 weeks) *Pirt*Cre;*Rosa26*Ai14 and *Pirt*Cre;Scn1a-floxed animals of both sexes were harvested on ice. For Tdtomato immunohistochemistry, spinal columns were fixed overnight at 4°C in 4% paraformaldehyde. For vesicular glutamate transporter 1 (VGLUT1) and choline acetyltransferase (ChAT) co-labeling experiments,

spinal columns were fixed in 4% paraformaldehyde for 1 hr on ice. Tissue was then placed in 30% sucrose solution overnight at 4°C. Following cryoprotection, tissue was embedded in optimal cutting temperature compound (OCT, Tissue-Tek Sakura) and stored at –80°C until sectioning. DRG from adult (8–15 weeks) animals of both sexes were harvested from thoracic spinal levels and fixed in 4% formaldehyde for 15 min at 4°C and were then incubated in 30% sucrose for 2–4 hr at 4°C. DRG were embedded in OCT and stored at –80°C until sectioning.

## Immunohistochemistry

Immunohistochemistry of spinal cord cryostat sections (30 μm) was performed using the following primary antibodies: rabbit anti-DsRed (1:3000, Takara Bio, 632496), guinea pig anti-VGLUT1 (1:8000, Zuckerman Institute, 1705), and rabbit anti-ChAT (1:10,000, Zuckerman Institute, 1574). Secondary antibodies used were as follows: anti-rabbit 594 (1:1000, Thermo Fisher, A32740), anti-guinea pig 488 (1:1000, Thermo Fisher, A11073), and anti-chicken 647 (Thermo Fisher, A32733). Specimens were mounted with Fluoromount-G with DAPI (SouthernBiotech, 0100-20). EDL muscles used in ex vivo muscle afferent recordings were placed in ice-cold 4% paraformaldehyde for 1 hr followed by ice-cold methanol for 15 min. Muscles were incubated in blocking solution (0.3% PBS-T and 1% BSA) followed by incubation in primary antibodies (guinea pig anti-VGLUT1 1:800 and chicken anti-NFH 1:300, Thermo Fisher ab4680) for 3–6 days at 4°C. After primary antibody treatment, tissue was washed in blocking solution and treated with secondary antibody (anti-guinea pig 488 1:50 and anti-chicken 594 1:300, Invitrogen, WA316328) for 2–3 days. Specimens were mounted with VECTASHIELD with DAPI (H-2000, Vector Laboratories). All specimens were imaged in three dimensions on a Zeiss LSM880 Airyscan confocal microscope. Images were analyzed using ImageJ software.

## Multiplex in situ hybridization

Fixed-frozen DRG tissue from 8- to 15-week-old mice of both sexes was cut into 25 μm sections and placed on electrostatically coated slides. Sections were processed for RNA in situ detection using a modified version of the manufacturer's protocol (Griffith et al., 2019, Advanced Cell Diagnostics) and the following probes: *Pvalb* (421931C1, mouse), *Runx3* (451271-C3, mouse), *Scn1a* (556181-C2, mouse), *Scn8a* (313341-C2, mouse), and *Scn9a* (434191-C2, mouse). Following in situ hybridization, sections were incubated in blocking solution (5% normal goat serum, 0.1% PBS-T) for 1 hr at room temperature (RT). Tissue was then incubated in primary antibodies overnight at 4°C. The following antibodies were used: rabbit DsRed (1:3000, Takara Bio, 632496), rabbit β3-tubulin (1:3000, Abcam, ab18207), chicken β3-tubulin (1:500, Abcam, ab41489), rabbit CGRP (1:1000, ImmunoStar, 24112), chicken GFP (1:3000, Abcam, ab13970), and chicken NFH (1:3000, Abcam, ab4680). Tissue was treated with the following secondary antibodies for 45 min at RT: anti-rabbit 448 (1:1000, Invitrogen, A32731), 594 (1:1000, Invitrogen, A11037) and 647 (1:1000, Invitrogen, A32733), anti-chicken 488 (1:1000, Invitrogen, A32931) and 594 (1:1000, Invitrogen, A32740). Sections were washed and mounted with Fluoromount-G with DAPI and imaged in three dimensions (2 μm axial steps) on an Olympus confocal (LV3000) using a ×40 0.90 NA water objective lens. Images were autothresholded and the probe signal integrated density for individual neurons was analyzed using ImageJ software. The coefficients of variation for *Scn1a*, *Scn8a*, and *Scn9a* integrated densities were calculated in Prism 9.0 (GraphPad Software) using the following formula: CV = $\mu/\sigma$ * 100.

## DRG culture preparation

DRGs were harvested from thoracic spinal levels of adult $Pvalb^{cre};Rosa26^{Ai14}$ and $Pirt^{cre};$Scn1a-floxed (6–16 weeks) mice of both sexes and transferred to $Ca^{2+}$-free and $Mg^{2+}$-free HBSS solution (Invitrogen, 14170-112). Upon isolation, processes were trimmed, and ganglia were transferred into collagenase (1.5 mg/mL; Type P, Sigma-Aldrich, 11213865001) in HBSS for 20 min at 37°C followed by TrypLE Express (Thermo Fisher, 12605-010) for 3 min with gentle rotation. TrypLE was neutralized with 10% horse serum (heat-inactivated; Invitrogen, 26050-070) and supplemented with culture media (MEM with L-glutamine, Phenol Red, without sodium pyruvate, Thermo Fisher, 11095-080), containing 10,000 U/mL penicillin-streptomycin (Thermo Fisher, 15140-122), MEM Vitamin Solution (Invitrogen, 11120-052), and B-27 supplement (Thermo Fisher, 17504-044). Serum-containing media was decanted and cells were triturated using a fire-polished Pasteur pipette in the MEM culture media described above. Cells were resuspended and triturated using a plastic pipette tip. Cells were plated on glass

coverslips that had been washed in 2 M NaOH for at least 4 hr, rinsed with 70% ethanol, UV-sterilized, and treated with laminin (0.05 mg/mL, Sigma-Aldrich, L2020-1MG) for 1 hr prior to plating. Cells were then incubated at 37°C in 5% $CO_2$. Cells were used for electrophysiology experiments 14–36 hr post-plating.

### In vitro electrophysiology

Whole-cell voltage-clamp recordings were made from dissociated DRG neurons using patch pipettes pulled from Model P-1000 (Sutter Instruments). Patch pipettes had a resistance of 3–5 MΩ when filled with an internal solution containing the following (in mM): 140 CsF, 10 NaCl, 1.1 EGTA, .1 $CaCl_2$, 10 HEPES, and 2.5 MgATP, pH with CsOH to 7.2. Seals and whole-cell configuration were obtained in an external solution containing the following (in mM): 145 NaCl, 5 KCl, 10 HEPES, 10 glucose, 2 $CaCl_2$, 2 $MgCl_2$, pH 7.3 with NaOH, osmolarity ~320 mOsm. Series resistance ranged from 6 to 11 MΩ and was compensated by 70–80%. Voltage errors were not directly assessed in current-clamp recordings. To isolate whole-cell sodium currents during voltage-clamp experiments, a modified external solution was applied containing the following (in mM): 15 NaCl, 130 TEA-Cl, 10 HEPES, 2 $BaCl_2$, 13 glucose, 0.03 $CdCl_2$, pH 7.3 with NaOH, osmolarity ~320 mOsm. Voltage-clamp recordings were performed at RT and current-clamp recordings were conducted at 37°C. Bath temperature was controlled and monitored using CL-100 (Warner Instruments).

### Ex vivo electrophysiology

The effect of the loss of $Na_V1.1$ on muscle spindle afferent firing rates during muscle stretch and sinusoidal vibration was determined using an isolated muscle nerve preparation. The extensor digitorum longus muscle and innervating peroneal branch of the sciatic nerve were dissected from adult (2–4-month-old) mice of both sexes. Muscles were held at optimal length (Lo), or the length of the muscle that maximal force of twitch contraction occurred. A series of nine 4 s ramp-and-hold stretches were given to three different stretch lengths repeated three times each (2.5, 5, and 7.5% Lo; ramp speed 40% Lo/s). A series of twelve 9 s sinusoidal vibrations were given (25, 50, and 100 μm amplitude; 10, 25, 50, and 100 Hz frequency). A 1-min rest was given between each length change. Firing rates during a 10 s baseline before stretch (resting discharge or RD) and the maximal firing rate during the rampup phase of stretch (dynamic peak or DP) were calculated for all animals. We determined whether the response to static stretch was maintained consistently throughout the 4 s stretch, as well as the coefficient of variability of the ISI during the plateau phase of stretch (CV = Std Dev/Mean of ISI over the time period of 1.5–3.5 s after end of rampup). Average firing rate during the 9 s of vibration and whether the unit could entrain in a 1:1 fashion to vibration was also determined. Detailed methods can be found in *Wilkinson et al., 2012*.

### Pharmacology

ICA 121431 (#5066), 4,9-anhydrotetrodotoxin (AH-TTX, #6159), and PF-05089771 (#5931) were purchased from Tocris Bioscience. Tetrodotoxin (TTX, ab120054) was purchased from Abcam. All other chemicals were from Sigma-Aldrich and Fisher Chemical.

### Data acquisition and analysis

Currents and voltages were acquired using pClamp software v11.2 (Molecular Devices). Recordings were obtained using an AxoPatch 200b patch-clamp amplifier and a Digidata 1550B and filtered at 5 kHz and digitized at 10 kHz. For biophysical analysis of whole-cell sodium currents, conductance (G) was calculated as $G = I/(V – E_{Na})$, in which $I$ is the peak current, $V$ is the voltage step, and $E_{Na}$ is the reversal potential for sodium calculated from Nernst equation based on the intracellular and extracellular sodium concentrations in our recording solutions (10.38 mV). Conductance data were normalized by the maximum conductance value, $G_{max}$, and data was fit with the Boltzmann equation: Fraction available = Minimum + ([Maximum-Minimum]/[1 + exp(V50 Vm)/$k$]), where $V_{50}$ denotes the membrane potential at which half the channels are inactivated and $k$ denotes the Boltzmann constant/slope factor. For voltage dependence of steady-state inactivation, peak current data were normalized based on the maximum current, $I_{max}$. Analysis of action potential amplitude, full-width half max, and threshold was performed on the first action potential elicited in response to a 100 pA current injection (100 ms). Action potential threshold was calculated as the membrane potential at which the first derivative

of the somatic membrane potential (dV/dT) reached 10 mV ms$^{-1}$ (*Griffith et al., 2019*; *Kress et al., 2008*). Tau values were calculated from 20 ms voltage steps from –90 mV to –30 mV and analyzed with single-exponential curve fits. Voltage-clamp and current-clamp experiments were analyzed with Clampfit software v11.2 (Molecular Devices) and custom MATLAB Scripts. Ex vivo recordings were obtained using an A-M Systems Model 1800 extracellular amplifier with headstage and digitized using an ADInstruments PowerLab. Data was analyzed using ADInstruments LabChart software using the Spike Histogram function.

### Experimental design and statistical analysis

Summary data are presented as mean ± SEM, from *n* cells, afferents, or sections, or *N* animals. For quantitative analysis of in situ hybridization data, at least three biological replicates per condition were used and the investigator was blinded to genotype for analysis. Behavioral experiments and analysis were also performed genotype-blind. Statistical differences were determined using parametric tests for normally distributed data and nonparametric tests for data that did not conform to Gaussian distributions or had different variances. Statistical tests are listed in 'Results' and/or figure legends. Statistical significance in each case is denoted as follows: *$p<0.05$, **$p<0.01$, ***$p<0.001$, and ****$p<0.0001$. Statistical tests and curve fits were performed using Prism 9.0 (GraphPad Software). All data generated or analyzed during this study are included in the article. Source data files have been uploaded to Mendeley for all figures (https://data.mendeley.com/datasets/kt23th75v9). Code has been uploaded to GitHub (https://github.com/darikoneil/PropAnalysisScripts; *O'Neil, 2022a*). A Key Resources Table with specific organism and reagent information has been included in the 'Materials and methods' section.

## Acknowledgements

This research was supported by a Postdoctoral Enrichment Program Award from the Burroughs Welcome Fund (TNG) and 5T32GM099608-10 and 1T32GM1144303-01A1 (CME). Work at San José State was supported by NIGMS 5SC3GM127195 (KAW) and NIGMS 5R25GM71381 (SO). Core facilities were supported by P30 EY12576. Part of this project was carried out at the Marine Biological Laboratory Neurobiology Course, with support from NINDS R25NS063307. Thanks to Drs. Jon Sack and Xinzhong Dong for sharing mouse lines, Drs. Jorge Contreras and Ioana Carcea for sharing behavioral equipment, Miguel Gonzalez Fernandez for writing code to automate ex vivo data analysis, and members of the Griffith laboratory for helpful discussions.

## Additional information

### Funding

| Funder | Grant reference number | Author |
|---|---|---|
| Burroughs Wellcome Fund | #1017423 | Theanne N Griffith |
| National Institute of General Medical Sciences | 5T32GM099608-10 | Cyrrus M Espino |
| National Institute of General Medical Sciences | 1T32GM1144303-01A1 | Cyrrus M Espino |
| National Institute of General Medical Sciences | 5SC3GM127195 | Katherine A Wilkinson |
| National Institute of General Medical Sciences | 5R25GM71381 | Serena Ortiz |
| National Institute of Neurological Disorders and Stroke | R25NS063307 | Kaylee M Wells Darik A O'Neil |

The funders had no role in study design, data collection and interpretation, or the decision to submit the work for publication.

## Author contributions
Cyrrus M Espino, Theanne N Griffith, Conceptualization, Resources, Data curation, Formal analysis, Supervision, Funding acquisition, Visualization, Methodology, Writing – original draft, Project administration, Writing – review and editing; Cheyanne M Lewis, Data curation, Formal analysis, Funding acquisition, Visualization, Writing – review and editing; Serena Ortiz, Data curation, Formal analysis, Visualization, Writing – review and editing; Miloni S Dalal, Kaylee M Wells, Data curation, Formal analysis, Writing – review and editing; Snigdha Garlapalli, Formal analysis, Investigation, Writing – review and editing; Darik A O'Neil, Resources, Data curation, Formal analysis, Writing – review and editing; Katherine A Wilkinson, Resources, Data curation, Formal analysis, Supervision, Funding acquisition, Methodology, Project administration, Writing – review and editing

## Author ORCIDs
Cyrrus M Espino (iD) http://orcid.org/0000-0003-2708-4577
Cheyanne M Lewis (iD) http://orcid.org/0000-0002-0057-2047
Katherine A Wilkinson (iD) http://orcid.org/0000-0002-2692-5533
Theanne N Griffith (iD) http://orcid.org/0000-0003-0090-6286

## Ethics
Animal use was conducted according to guidelines from the National Institutes of Health's Guide for the Care and Use of Laboratory Animals and was approved by the Institutional Animal Care and Use Committee of Rutgers University-Newark (PROTO201900161), UC Davis (#21947 and #22438) and San José State University (#990, ex vivo muscle recordings).

## Decision letter and Author response
Decision letter https://doi.org/10.7554/eLife.79917.sa1
Author response https://doi.org/10.7554/eLife.79917.sa2

## Additional files

### Supplementary files
• MDAR checklist

### Data availability
Source Data files have been uploaded to Mendeley for all figures (https://data.mendeley.com/datasets/kt23th75v9). Code has been uploaded to GitHub (https://github.com/doctheagrif/Current-Clamp-Matlab-Code_O-Neil-DA, copy archived at swh:1:rev:12667fd74eb68a8cd7ec858f33d9fec6e5c46772). A key resources table with specific organism and reagent information has been included in the methods section.

The following dataset was generated:

| Author(s) | Year | Dataset title | Dataset URL | Database and Identifier |
|---|---|---|---|---|
| Griffith T | 2022 | Nav1.1 is essential for proprioceptive signaling and motor behaviors | https://doi.org/10.17632/kt23th75v9.3 | Mendeley Data, 10.17632/kt23th75v9.3 |

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
