## [Editor Report]

This article provides insight into the importance of a voltage-gated sodium channel in proprioceptors, a group of mechanosensory neurons that target muscle. Using pharmacology, gene knockout, behavior, and histology in mice, the authors show quite convincingly that Na_V_1.1 in sensory neurons is essential for normal motor behavior and contributes to proprioceptor excitability. The work has interesting implications for human subjects with inherited variants of Na_v_1.1.

---

## [Decision Letter]

**Decision letter after peer review:**

Thank you for submitting your article "Nav1.1 in mammalian sensory neurons is required for normal motor behaviors" for consideration by *eLife*. Your article has been reviewed by 3 peer reviewers, including Alexander Theodore Chesler as Reviewing Editor and Reviewer #1, and the evaluation has been overseen by Richard Aldrich as the Senior Editor. The following individual involved in the review of your submission has agreed to reveal their identity: Joriene C de Nooij (Reviewer #2).

Overall, all the reviewers agree this is a well-executed and interesting study. Each reviewer raised specific points that needed clarification (see below). While the majority can be addressed with revised text, a few key experiments are being requested that seem reasonable, would add a great deal, and therefore should be done where possible. Please provide a point-by-point response to each comment when you resubmit.

*Reviewer #1 (Recommendations for the authors):*

1. Figure 2 needs a proper drug-free control. The authors should show at least a few recordings demonstrating the effect of perfusing vehicles on the sodium currents and excitability of the proprioceptors. I don't doubt the effect of ICA is real, but this seems like a basic requirement for this kind of experiment.

2. "In voltage clamp experiments, ICA application reduced the whole-cell sodium current (INa) on average by 46%, suggesting NaV1.1 is a dominant sodium channel isoform in these neurons (Figure 2A-B)." This statement is misleading. In principle, the other 54% might be mediated entirely by NaV1.6 or NaV1.7. Very good selective blockers of NaV1.7 (PF-771) are commercially available, and reasonably good NaV1.6 blockers (4,9-AH-TTX), thus defining which channels carry the rest of the current in proprioceptors is a doable experiment. Given the current clamp experiments found no effects of NaV1.1. block on excitability in 25% of proprioceptors, it is likely the other A fiber enriched sodium channels are playing a major role. NaV1.6 especially appears to be critical for the function of myelinated DRG afferents.

3. It's surprising the authors generated a conditional NaV1.1. KO mouse and performed no in vitro recordings on DRGs from these mice. I understand their current cross provides no means of labelling proprioceptors for targeted patching, and brain recombination of PV-Cre precludes using this line for behavior. However, if the Parv-Cre X Nav1.1 KO mice are viable, then recording from DRGs in vitro should be possible. These neurons could be labeled with Cre-dependent AAVs for targeted recordings (e.g. flex-tdTomato). It seems important to validate that ICA-sensitive current is gone in large diameter DRGs in their conditional KO, which would nicely validate both the efficiency of the KO and the concentration of ICA they used.

4. Considering the behavior reported, it is likely that the effect of Nav1.1 might be on golgi tendon organs (GTOs) rather than spindles (see Akay et al. 2014). The fact that the vibratory muscle responses are intact, but the stretch (which would also activate the GTOs) are affected is consistent with this conclusion. Verifying the structure of the GTOs would be helpful.

5. One important consideration is that sodium channel α subunit deletion in DRGs has been reported to produce compensatory upregulation of other isoforms (e.g. NaV1.8 deletion produces compensatory upregulation of NaV1.7, in Akopian 1999). This is a strong argument for focusing their in vitro recording on selectively pharmacological approaches, as opposed to the conditional KO, and should be mentioned. Genetic compensation might also explain the contrast between the dramatic effects of ICA on excitability measured in vitro, versus the weaker effects of the conditional KO on excitability measured in vivo.

6. It is interesting the authors find behavioral effects of even heterozygous deletion of NaV1.1. If NaV1.6 accounts for 46% of the total sodium current in proprioceptors, then it's impressive that removal of just 23% of the total sodium current can produce behavioral and static force firing alterations (and presumably change in vitro excitability). Honestly, this result is quite surprising as most DRG sodium channels are not haploinsufficient (as shown with conditional and global deletion of NaV1.7, NaV1.8, NaV1.9, and NaV1.6, and in human null mutants for NaV1.7 and NaV1.9) and suggests that proprioceptors are unique in their non-redundant dependence on NaV1.1.

7. Staining 1a afferents with Vglut1 should be combined with Parvalbumin. The finding that motoneurons (which to be proper should be called ventrally located cholinergic neurons) have ~ 8 Vglut1 terminals in WT versus ~ 5 in cKO seems a rather small difference. To make claims about connectivity/spinal effects would require monitoring monosynaptic reflex amplitude evoked by stimulating the dorsal root.

8. This brings us to the last point, which is probably the most important in terms of the conclusions of the paper, but the experiment is probably not needed for publication, just interesting. Conditional KO of NaV1.1 produces such strong behavioral effects on motor function, and NaV1.1 block so profoundly impairs proprioceptor firing in vitro, that it's quite surprising that the effect of KO on the in vivo firing is either modest (for static) or nonexistent (for dynamic). It's thus really interesting the authors find by IHC a reduced anatomical synaptic input to motor neurons. This provokes the obvious hypothesis that the behavioral deficits result from both decreased peripheral excitability and deficits at the central terminal of the proprioceptor (as mentioned in the discussion has been shown for NaV1.7 in both nociceptors and olfactory sensory neurons). If NaV1.1. is the predominant sodium channel in proprioceptors, then it is likely it is expressed all along the proprioceptor neurons, and may contribute to both AP initiation, propagation, invasion of the presynaptic terminal, and presynaptic excitability. Expression of ChR2 in NaV1.1-deleted sensory neurons would enable a physiological study of how NaV1.1 KO affects synaptic transmission between sensory afferents and proprioceptor targets in the spinal cord. A better (and easier experiment) would be to inject an AAV-encoding ChR2 into PV-Cre mice at P0, and then record light-evoked EPSCs in proprioceptor target neurons in the spinal cord while applying ICA onto the spinal cord slice. The advantage of the pharmacological experiment is it would allow the authors to disambiguate the anatomical loss of connectivity due to conditional KO, from any intrinsic effects of NaV1.1 block on a presynaptic function.

*Reviewer #2 (Recommendations for the authors):*

The authors appear extremely careful not to overstate their findings in the title and abstract – perhaps overly so in certain cases. For instance, I was surprised to find the selective requirement of Nav1.1 only in the data section of the ex vivo muscle spindle analyses. Bringing in these observations earlier (in the abstract) may make the manuscript less generic and more enticing for a number of people. In part, the careful wording may be in recognition of the limitations of some of the experiments – which in my few could be helped by adding a few additional experiments/controls (specific recommendations below). Other recommendations concern the presentation of the data in some of the Figures.

1. Three comments on the RNAscope in situ data. (A) Figure 1 nicely demonstrates the distribution of Nav1.1 across various sensory subsets and in Figure 6 it is used to demonstrate normal numbers of proprioceptors across all genotypes. As part of these analyses, it also shows the expression of B3 tubulin. The B3 tubulin is not helping in the interpretation of these data. I'd strongly suggest taking this data out and just showing the merged image with Nav1.1 and the marker gene of interest (if insisting on keeping the B3 panel does the merge without B3 tubulin). (B) In Figure 1, the summary plots for CGPR and MrgprD are a bit confusing; it would be better to place the % next to the pink segments rather than in the yellow (non-overlapping) segments. C) It is mentioned that "the coefficient of variation in integrated fluorescence density in proprioceptors is 76.5" (lines 253-255 and Figures 1E and F). The significance of this statement, or how one derives this number is not explained in the text or methods.

2. In probing the role of Nav1.1 in proprioceptors using in vitro recordings it may be useful for the reader to get a sense of the expression (levels) of all Nav channels in proprioceptors in order to better understand how blocking or activating Nav1.1 may affect neuronal excitability. Currently, only Nav1.3 is mentioned in the context of the Nav1.1 blocker ICA. Nav1.3 is not considered to play a role because it is not known to be expressed in proprioceptors, and because "those channels are not expressed in uninjured adult DRG neurons". The latter statement is a bit odd in this context given that dissociated DRG neurons may well be considered injured. It may be useful to test the Nav1.3 transcript levels in the dissociated tdT positive neurons (after culturing) to exclude the possibility this channel interferes with the interpretation of these studies.

3. "inhibition of NaV1.1 significantly reduced the number of evoked action potentials in most genetically identified proprioceptors (Figure 2H); however, of the 20 cells recorded, 5 had low firing rates that were not further inhibited by ICA, indicating in some proprioceptors subtypes other NaVs mediate (to) action potential firing." I am a bit puzzled by Figure 2H, as the average for the ICA data seems to align with the top most data point? Is this correct? From this statement, it also becomes clear that Nav1.1 may be more relevant in certain proprioceptor subtypes than in others, which is an interesting observation that could warrant more discussion.

4. The in vitro recordings suggest a role for Nav1.1 in proprioceptor repetitive firing, yet these experiments would carry so much more weight if performed with Nav1.1 mutant (or even Nav1.1. heterozygous) proprioceptors. While the behavioral studies with PVCre are understandably not informative, the PVcre-induced conditional Nav1.1 deletion (in conjunction with the tdT reporter) could have been used to solidify the in vitro studies. Given that all the genetic tools are in place, it is not clear to me why this was not attempted or discussed.

5. The behavioral analyses of Nav1.1cKO animals show a clear loss of motor coordination, and mutant animals differ significantly from wild-type (Nav1.1flx/flx) controls in open field and rotarod. While both behaviors may also rely on low threshold cutaneous afferent feedback (in particular in rotarod assays) which similarly expresses Nav1.1, it is likely that the phenotype is in part due to the loss in proprioceptors. As a control, the authors assess the expression of the Pirt:Cre driver in the spinal cord (using a tdT reporter). For completion, it would also be informative to perform similar control studies in muscle (e.g. intrafusal muscle fibers) brain stem, cerebellum, and cortex. Kim et al. demonstrated little to no expression in these tissues, yet it is not clear whether those data also apply to adult animals.

6. The ex vivo muscle spindle recordings offer a nice complement to the in vitro and behavioral studies and permit a more specific assessment of a key subset of proprioceptors, the muscle spindle afferents. These data are interesting for two reasons, the similarity in phenotype between the cHET and cKO animals (i.e. the haploinsufficiency of Nav1.1), and the apparent selective requirement of Nav1.1 for the static MS response. Yet these studies also demonstrate the variability in the data, perhaps suggesting that the number of recordings may need to be higher? As for the selective requirement of Nav1.1 in static discharge frequency, it is not clear to me why this is not more highlighted in the abstract. This is an interesting result that may help to tease apart the molecular basis of the response properties of different proprioceptor subtypes.

7. Figure 4 panels A, B, and C are a bit confusing with regard to the pie charts. The summary quantifications for static discharge patterns for wt, het, and cKO Nav1.1. mice are depicted with the amount of 'inconsistent traces' in black. This color coding is reversed for the resting discharge (absent discharge is quantified as colored; even though in wt 50% of afferents lack resting discharge, this % is higher in the mutants).

8. In a set of control studies, the Nav1.1 genotypes are compared with respect to the number of proprioceptors, spindle morphology, and synaptic connectivity between MS afferents and spinal motor neurons. Interestingly, these studies appear to reveal a significant reduction in MS afferent vGlut1 positive synaptic contacts onto ChAT positive motor neurons. This is another potentially interesting result but these data appear a bit rushed. SN-MN connectivity is best compared across the same (backfilled) motor pool (and ideally by only looking at synapses from the homonymous afferents). Another aspect that may be influencing these data is the size of the synaptic boutons on the MNs. Maybe this can be resolved using higher power images (again comparing the contacts on motor neurons that innervate the same muscle)? Ultimately, the best assessment of these connections is to perform ventral root recordings after stimulating the dorsal roots. (FWIW, the DAPI panels in Figure 7 are more distractive than useful; an orientating schematic may also help).

9. There are various other studies that have performed in vitro recordings on proprioceptor afferents with sometimes divergent observations (e.g. Woo et al., 2015; Florez Paz et al., 2016; Zheng, et al., 2019; Oliver et al., 2021). It would have been interesting to see a discussion of the current observations in the context of these other studies.

*Reviewer #3 (Recommendations for the authors):*

This is a very thorough study on Nav1.1 function in the peripheral nervous system. The data are generally of high quality, but I have some questions about the patch-clamp data:

– Voltage-gated sodium channels are very fast gating, therefore it is crucial to have good space and voltage-clamp conditions for their measurement. Pipette resistances up to 6Mohm, as used here, are very high, and it is quite likely that the currents recorded by the authors were not well clamped. The quality of the clamp can be assessed by the shape of the current traces, thus the authors should show example IV traces, which allow judgement of the current at low voltages, where the driving force is high, but the open probability of the channels is low. This should be added to Figure 2.

– Figure 2: As stated in the first point, the quality of the presented data is hard to assess. Please show the stimulation protocols. Please show a complete IV curve. At which potential was current decay assessed? Why not over a range of potentials, which would give a better understanding of the impact during an AP. Time to peak – in your case called rise time, is very fast in Navs. Again, clamping conditions are crucial. It seems that in the ICA group there is an outlier, which most likely has a higher series resistance (how high was it in general?) and thus should be removed. If done so, it is quite likely, that there will be no significant difference anymore.

– The blocker used: ICA is also effective against Nav1.3 as mentioned in the manuscript. Please show that your cultures do not express NAv1.3 or only to a minor degree. Dissociation is quite an axotomy and may thus induce Nav1.3 expression in mice. Alternatively, use cells that are unlikely to express Nav1.1 as a control for your blocker, such as MrgprD+ neurons, which you investigated in Figure 1.

– Discussion lines 586ff: it is easy to investigate the persistent current in your recordings. Please do so, so you can substantiate your interpretation and discussion of this point. Also, using the correct protocol it is possible to investigate if the proprioceptors show resurgent currents. An in-depth investigation of the biophysics of Nav1.1 in their expressing cells (i.e. proprioceptors) would strengthen the manuscript a lot. It could also help to understand why the channel specifically is so important for proprioceptor function, while the other channels, which are expressed in these cells, cannot compensate for it. How quickly is the Nav1.1 component recovering in proprioceptors? is this really the important characteristic which is missing in the ko mice?

– Figure 1: please explain E and F better – what is plotted on the x-axis? What is the value 76.5 mentioned in the text in line 255?

– I am a bit confused by your statement in line 293 on the Vhalf of activation of Nav1.1 being -15mV. you would at least need to compare it to the Vhalf of the other Navs in order to make a statement on their involvement in the threshold. How did you determine this relatively negative threshold? In general, please provide more info on your ephys analysis (was tau obtained by single expt fitting? did you include an offset, was this changed by ICA?).

– Suppl Figure 2: which mouse is this? it is good to see the expression also in the ventral horn, please also show a 40x of the ventral horn.

– Please define Lo also in the Figure legend or in the text. Figure 4: what does the percentage during the resting state refer to? fibers – time recorded?

– Fi.5A-c: how did you test for differences in the percentage of entrainment? cKO has 64% vs fl/fl 33% – this seems to be quite a difference (in contrast to your statement in line 433 where you describe them to be similar).

– Figure 5D: what does FPS stand for?

– Figure 8: you mention Nav1.6 and Nav1.7 in your figure. Are they expressed in your RNAscope data in proprioceptors? Do you have any other data on other Nav subtypes?

– Discussion: it would be nice to mention that Nav channels are also mechanosensitive, with potential differences among sodium channel subtypes. This could contribute to their function in proprioceptors.

---

## [Author Response]

Reviewer #1 (Recommendations for the authors):1. Figure 2 needs a proper drug-free control. The authors should show at least a few recordings demonstrating the effect of perfusing vehicles on the sodium currents and excitability of the proprioceptors. I don't doubt the effect of ICA is real, but this seems like a basic requirement for this kind of experiment.

Thanks to the Reviewer for this suggestion. We have added this experiment to a new Figure 2—figure supplement 1, in which we find no significant effect of DMSO on proprioceptor sodium currents.

2. "In voltage clamp experiments, ICA application reduced the whole-cell sodium current (INa) on average by 46%, suggesting NaV1.1 is a dominant sodium channel isoform in these neurons (Figure 2A-B)." This statement is misleading. In principle, the other 54% might be mediated entirely by NaV1.6 or NaV1.7. Very good selective blockers of NaV1.7 (PF-771) are commercially available, and reasonably good NaV1.6 blockers (4,9-AH-TTX), thus defining which channels carry the rest of the current in proprioceptors is a doable experiment. Given the current clamp experiments found no effects of NaV1.1. block on excitability in 25% of proprioceptors, it is likely the other A fiber enriched sodium channels are playing a major role. NaV1.6 especially appears to be critical for the function of myelinated DRG afferents.

We appreciate the Reviewer’s suggestions to examine the function of the other Na_V_ isoforms in proprioceptors. We agree that this analysis adds value to our study. We have now performed pharmacological experiments to identify the contributions of all Na_V_ channels in proprioceptors to the whole-cell sodium current using in vitro patch-clamp electrophysiology. The results (Figure 2E-F) show that Nav1.6 and Nav1.7 make up 14.5% and 32.9% of the somal sodium current in proprioceptors, respectively. We did find that on average our pharmacological strategy left ~7.8% of the sodium current unblocked; however, that portion was TTX sensitive, suggesting imperfect block the antagonists used, which we addressed in the main text.

3. It's surprising the authors generated a conditional NaV1.1. KO mouse and performed no in vitro recordings on DRGs from these mice. I understand their current cross provides no means of labelling proprioceptors for targeted patching, and brain recombination of PV-Cre precludes using this line for behavior. However, if the Parv-Cre X Nav1.1 KO mice are viable, then recording from DRGs in vitro should be possible. These neurons could be labeled with Cre-dependent AAVs for targeted recordings (e.g. flex-tdTomato). It seems important to validate that ICA-sensitive current is gone in large diameter DRGs in their conditional KO, which would nicely validate both the efficiency of the KO and the concentration of ICA they used.

We appreciate the Reviewer’s suggestion. We chose to perform ex-vivo recordings from muscle-spindle afferents instead of somal recordings from Na_V_1.1^cko^ mice. Unfortunately, PV^Cre^;Nav1.1^fl/fl^ mice die prematurely and have severe seizures (PMID: 17537961), precluding their use in this study. The concentration of ICA used is in line we our previously published data and that of others (PMIDs: 31300524, 28607094). As a control, we performed ICA recordings in our knockout animals (Figure 4 —figure supplement 1) and surprisingly found a small but significant decrease in the current when recording from large-diameter neurons, indicative of upregulated Na_V_1.3 channels as suggested by Reviewers 2 and 3. We have addressed this in the main text and adjusted our interpretation of pharmacological data accordingly.

Results:

“It is important to note that ICA 121431 also blocks Na_V_1.3 channels, which could be upregulated in our cultured DRG neuron preparations (Wangzhou et al., 2020). Indeed, a small but significant decrease in I_Na_ was observed in recordings from large-diameter DRG neurons harvested from Pirt^Cre^;Na_V_1.1^fl/fl^ mice (Na_V_1.1^cKO^), which lack Na_V_1.1 in all sensory neurons. Thus, inhibition of upregulated Na_V_1.3 channels could contribute to the effect of on ICA on the proprioceptor I_Na_.”

Discussion:

“These results, however, should be interpreted with the caveat that in these experiments ICA may also be acting on Na_V_1.3 channels that are upregulated during DRG neuron culturing.”

4. Considering the behavior reported, it is likely that the effect of Nav1.1 might be on golgi tendon organs (GTOs) rather than spindles (see Akay et al. 2014). The fact that the vibratory muscle responses are intact, but the stretch (which would also activate the GTOs) are affected is consistent with this conclusion. Verifying the structure of the GTOs would be helpful.

We agree with the Reviewer that there are likely alterations in GTO afferent function as well. We are confident that the afferents we record from in the ex vivo system are muscle spindle afferents, since we have never seen increased firing during twitch contraction that would be typical of a Ib afferent. However, it is formally possible that we have included some Ib afferents in our sample. We think the unaltered dynamic sensitivity finding would likely hold even if that were the case, because Dynamic Peak firing during stretch was also unaltered. We also predict that GTO structure is likely to be preserved in our mice, although we don’t currently have a good method to quantify that. We have added the following to our discussion to highlight the fact that we did not assay GTO function but would predict it is likely affected.

Discussion:

“Due to the ubiquitous expression of Na_V_1.1 in all proprioceptors and the importance of Golgi Tendon Organ feedback to motor control, alterations in function in those proprioceptors likely contribute to the behavioral deficits we observed. However, we did not directly measure their function.”

5. One important consideration is that sodium channel α subunit deletion in DRGs has been reported to produce compensatory upregulation of other isoforms (e.g. NaV1.8 deletion produces compensatory upregulation of NaV1.7, in Akopian 1999). This is a strong argument for focusing their in vitro recording on selectively pharmacological approaches, as opposed to the conditional KO, and should be mentioned. Genetic compensation might also explain the contrast between the dramatic effects of ICA on excitability measured in vitro, versus the weaker effects of the conditional KO on excitability measured in vivo.

We thank this reviewer for highlighting this point. We have this information currently in the Discussion section:

“The lack of an effect on dynamic sensitivity could suggest the upregulation of other Na_V_ subtypes or other molecules as a compensatory mechanism to counteract the loss of Na_V_1.1.”

We have also noted that in vitro recordings are performed from the soma, whereas ex vivo recordings are done from axons. This could also explain the differential affects between recording strategies and may suggest different functional contributions of Na_V_1.1 to somal versus afferent firing.

“Indeed, our in vitro electrophysiological experiments found a more pronounced effect of acute Na_V_1.1 inhibition on proprioceptor excitability. This could be due, however, to artificially upregulated Na_V_1.3 channel activity in culturing conditions, or conversely, a higher density of Na_V_1.1 expression in proprioceptor somata. Future studies using temporally controlled deletion of Na_V_1.1 in sensory neurons could tease this apart.”

6. It is interesting the authors find behavioral effects of even heterozygous deletion of NaV1.1. If NaV1.6 accounts for 46% of the total sodium current in proprioceptors, then it's impressive that removal of just 23% of the total sodium current can produce behavioral and static force firing alterations (and presumably change in vitro excitability). Honestly, this result is quite surprising as most DRG sodium channels are not haploinsufficient (as shown with conditional and global deletion of NaV1.7, NaV1.8, NaV1.9, and NaV1.6, and in human null mutants for NaV1.7 and NaV1.9) and suggests that proprioceptors are unique in their non-redundant dependence on NaV1.1.

We thank the reviewer for this comment. We also found this finding very intriguing and surprising and have included extra discussion of this point:

“Despite this limitation, one noteworthy and intriguing finding from our study was the haploinsufficiency of Na_V_1.1 in sensory neurons for proprioceptor function and normal motor behavior in the open field test. At the afferent level, heterozygous and homozygous loss of Na_V_1.1 produced similar deficits in static firing, suggesting that loss of less than a quarter of the proprioceptor I_Na_ is sufficient to impair proprioceptor responsiveness to muscle stretch. Na_V_1.1 is haploinsufficeint in several brain neuron cell-types for normal excitability and function, suggesting the contributions of Na_V_1.1 to neuronal function are highly sensitive to genetic perturbations.”

7. Staining 1a afferents with Vglut1 should be combined with Parvalbumin. The finding that motoneurons (which to be proper should be called ventrally located cholinergic neurons) have ~ 8 Vglut1 terminals in WT versus ~ 5 in cKO seems a rather small difference. To make claims about connectivity/spinal effects would require monitoring monosynaptic reflex amplitude evoked by stimulating the dorsal root.

We thank reviewer for this comment. We re-performed our spinal cord experiments and analyses. We were unable to label proprioceptive afferents with our parvalbumin antibody and only observed interneuron staining (see Author response image 1). This appears consistent with immunohistochemical staining from other groups using adult mice (PMIDs: 12457066, 24652212). To analyze proprioceptive axon targeting to motor neurons more rigorously, we only quantified puncta on ChAT positive motor neurons that were over 400 µm^2^ (ref). Using this criterion, we now find no significant differences between genotypes. It is possible that we were including γ motor neurons in our previously analysis and thank the reviewer for bringing this to our attention. These updated experiments are now in Figure 7. We have also modified our model figure and the main text.

**Author response image 1. sa2fig1:** Spinal cord of a Na_V_1. 1f^l/fl^ mouse stained with an anti-parvalbumin antibody (Thomas Jessell Laboratory, Columbia University). Strong labeling is seen in PV+ interneurons, but not in Proprioceptive afferent terminals.

8. This brings us to the last point, which is probably the most important in terms of the conclusions of the paper, but the experiment is probably not needed for publication, just interesting. Conditional KO of NaV1.1 produces such strong behavioral effects on motor function, and NaV1.1 block so profoundly impairs proprioceptor firing in vitro, that it's quite surprising that the effect of KO on the in vivo firing is either modest (for static) or nonexistent (for dynamic). It's thus really interesting the authors find by IHC a reduced anatomical synaptic input to motor neurons. This provokes the obvious hypothesis that the behavioral deficits result from both decreased peripheral excitability and deficits at the central terminal of the proprioceptor (as mentioned in the discussion has been shown for NaV1.7 in both nociceptors and olfactory sensory neurons). If NaV1.1. is the predominant sodium channel in proprioceptors, then it is likely it is expressed all along the proprioceptor neurons, and may contribute to both AP initiation, propagation, invasion of the presynaptic terminal, and presynaptic excitability. Expression of ChR2 in NaV1.1-deleted sensory neurons would enable a physiological study of how NaV1.1 KO affects synaptic transmission between sensory afferents and proprioceptor targets in the spinal cord. A better (and easier experiment) would be to inject an AAV-encoding ChR2 into PV-Cre mice at P0, and then record light-evoked EPSCs in proprioceptor target neurons in the spinal cord while applying ICA onto the spinal cord slice. The advantage of the pharmacological experiment is it would allow the authors to disambiguate the anatomical loss of connectivity due to conditional KO, from any intrinsic effects of NaV1.1 block on a presynaptic function.

We thank the Reviewer for these excellent suggestions for future experiments. Indeed, we are planning to use optogenetics to tease out the contributions of Na_V_1.1 to mechanically induced signaling versus axonal excitability. While we no longer observe significant changes in proprioceptor axon targeting of motor neurons, in future experiments, we will look more specifically at potential central deficits in our model, including homonymous vs. heteronymous sensory-motor connectivity, whose balance was previously found to be affected by loss of proprioceptor activity (PMID: 26094608). We also plan to perform circuit tracing and functional spinal cord electrophysiological experiments as we continue to build this new and exciting line of research in our lab.

Reviewer #2 (Recommendations for the authors):The authors appear extremely careful not to overstate their findings in the title and abstract – perhaps overly so in certain cases. For instance, I was surprised to find the selective requirement of Nav1.1 only in the data section of the ex vivo muscle spindle analyses. Bringing in these observations earlier (in the abstract) may make the manuscript less generic and more enticing for a number of people. In part, the careful wording may be in recognition of the limitations of some of the experiments – which in my few could be helped by adding a few additional experiments/controls (specific recommendations below). Other recommendations concern the presentation of the data in some of the Figures.1. Three comments on the RNAscope in situ data. (A) Figure 1 nicely demonstrates the distribution of Nav1.1 across various sensory subsets and in Figure 6 it is used to demonstrate normal numbers of proprioceptors across all genotypes. As part of these analyses, it also shows the expression of B3 tubulin. The B3 tubulin is not helping in the interpretation of these data. I'd strongly suggest taking this data out and just showing the merged image with Nav1.1 and the marker gene of interest (if insisting on keeping the B3 panel does the merge without B3 tubulin). (B) In Figure 1, the summary plots for CGPR and MrgprD are a bit confusing; it would be better to place the % next to the pink segments rather than in the yellow (non-overlapping) segments. (C) It is mentioned that "the coefficient of variation in integrated fluorescence density in proprioceptors is 76.5" (lines 253-255 and Figures 1E and F). The significance of this statement, or how one derives this number is not explained in the text or methods.

We thank the reviewer for these comments that improve figure readability. We have made the following changes to Figure 1:

a) Thank you for this comment, we have removed the tubulin staining from these figures.

b) We have changed the location of the percentages and colored them magenta.

c) The coefficient of variation is used to describe variability in data, and we have added more information about this metric to the main text. With new RNAscope data added for Na_V_1.6 and Na_V_1.7 in proprioceptors, we now compare coefficient of variations between Na_V_ isoforms and include this in our discussion of potential differences in Na_V_ expression in distinct proprioceptor subtypes.

“This was quantified using the coefficient of variation, a relative measure of the extent of variations within data. The coefficient of variation for Na_V_1.1 expression was calculated to be 75.6, whereas this value increased to 97.3 and 88.1 for Na_V_1.6 and Na_V_1.7, respectively. This indicates that while all three isoforms are ubiquitously expressed in proprioceptors, the relative levels differ, with Na_V_1.1 having the most consistent level of expression across neurons analyzed. Furthermore, the average integrated density of the Na_V_1.1 signal for a given proprioceptive DRG neuron was significantly higher than both Na_V_1.6 and Na_V_1.7 (Figure 2E, p < 0.0001).”

2. In probing the role of Nav1.1 in proprioceptors using in vitro recordings it may be useful for the reader to get a sense of the expression (levels) of all Nav channels in proprioceptors in order to better understand how blocking or activating Nav1.1 may affect neuronal excitability. Currently, only Nav1.3 is mentioned in the context of the Nav1.1 blocker ICA. Nav1.3 is not considered to play a role because it is not known to be expressed in proprioceptors, and because "those channels are not expressed in uninjured adult DRG neurons". The latter statement is a bit odd in this context given that dissociated DRG neurons may well be considered injured. It may be useful to test the Nav1.3 transcript levels in the dissociated tdT positive neurons (after culturing) to exclude the possibility this channel interferes with the interpretation of these studies.

We thank the reviewer for this suggestion. We have now performed RNAscope analysis of both Na_V_1.6 and Na_V_1.7 expression in proprioceptors, as well as functional pharmacological experiments using in vitro electrophysiology (see comments to Reviewer #1). These new results are found in Figure 2. Additionally, as mentioned above, we performed ICA recordings on large-diameter neurons from Na_V_1.1^cko^ animals and found a small but significant effect, suggesting either non-specific actions on native sodium channels or upregulated N_aV_1.3. We have discussed this caveat in the main text and adjusted interpretations in the Discussion. Nevertheless, we do not think it changes our overall conclusions given ex-vivo recordings and our behavioral data.

3. "inhibition of NaV1.1 significantly reduced the number of evoked action potentials in most genetically identified proprioceptors (Figure 2H); however, of the 20 cells recorded, 5 had low firing rates that were not further inhibited by ICA, indicating in some proprioceptors subtypes other NaVs mediate (to) action potential firing." I am a bit puzzled by Figure 2H, as the average for the ICA data seems to align with the top most data point? Is this correct? From this statement, it also becomes clear that Nav1.1 may be more relevant in certain proprioceptor subtypes than in others, which is an interesting observation that could warrant more discussion.

We have fixed this error in Figure 2H, which occurred during the overlay of the two graphs. We thank the reviewer for drawing our attention to this mistake.

4. The in vitro recordings suggest a role for Nav1.1 in proprioceptor repetitive firing, yet these experiments would carry so much more weight if performed with Nav1.1 mutant (or even Nav1.1. heterozygous) proprioceptors. While the behavioral studies with PVCre are understandably not informative, the PVcre-induced conditional Nav1.1 deletion (in conjunction with the tdT reporter) could have been used to solidify the in vitro studies. Given that all the genetic tools are in place, it is not clear to me why this was not attempted or discussed.

As mentioned in comments to Reviewer #1, we focused our electrophysiological analysis of Na_V_1.1^cko^ proprioceptors on ex vivo experiments, as these allow us to record directly from axons as opposed to the soma. in vitro recordings from DRG somata are an excellent way to determine channel functionality in genetically identified neurons, as well as understand how channels may shape sensory neuron intrinsic excitability at the biophysical level. It is unclear, however, what the contribution of channels expressed on DRG soma is to the actual transmission of somatosensory signals. We are currently establishing techniques in the lab that will allow for biophysical analyses of sodium currents in our models, which we will leverage in future experiments.

5. The behavioral analyses of Nav1.1cKO animals show a clear loss of motor coordination, and mutant animals differ significantly from wild-type (Nav1.1flx/flx) controls in open field and rotarod. While both behaviors may also rely on low threshold cutaneous afferent feedback (in particular in rotarod assays) which similarly expresses Nav1.1, it is likely that the phenotype is in part due to the loss in proprioceptors. As a control, the authors assess the expression of the Pirt:Cre driver in the spinal cord (using a tdT reporter). For completion, it would also be informative to perform similar control studies in muscle (e.g. intrafusal muscle fibers) brain stem, cerebellum, and cortex. Kim et al. demonstrated little to no expression in these tissues, yet it is not clear whether those data also apply to adult animals.

We thank the reviewer for this suggestion. If our Pirt^Cre^ driver were hitting the brain regions mentioned, our mice would develop severe epilepsy and would not be viable in adulthood. Thus, we are confident that our expression is limited to the peripheral nervous system, as previously published.

6. The ex vivo muscle spindle recordings offer a nice complement to the in vitro and behavioral studies and permit a more specific assessment of a key subset of proprioceptors, the muscle spindle afferents. These data are interesting for two reasons, the similarity in phenotype between the cHET and cKO animals (i.e. the haploinsufficiency of Nav1.1), and the apparent selective requirement of Nav1.1 for the static MS response. Yet these studies also demonstrate the variability in the data, perhaps suggesting that the number of recordings may need to be higher? As for the selective requirement of Nav1.1 in static discharge frequency, it is not clear to me why this is not more highlighted in the abstract. This is an interesting result that may help to tease apart the molecular basis of the response properties of different proprioceptor subtypes.

We agree with the reviewer that our ex vivo data highlight two very important findings and we have revised our abstract to bring the more exciting findings in our study to the forefront. We also agree that there is variability in the response, with some afferents appearing to respond to static stretch normally and at least half of the afferents lacking one or both copies of Na_V_1.1 showing inconsistent firing during stretch. Even in wild type animals, we do see a fair amount of variability in the relative static and dynamic response to stretch, although we almost never see the sort of inconsistent firing observed here (e.g. Wilkinson, et al., PLoS One, 2012). Since we cannot functionally identify Ia vs II responses in mice, it is possible that some of the variability is due to Na_V_1.1 playing a different or more important role in different subtypes of afferents. However, the relatively consistent expression of Nav1.1 in proprioceptor soma may suggest this is not the case. More likely, there could be differences in protein expression levels and/or some developmental compensation to the loss of Na_V_1.1. This is a question we hope to address in future studies using induced deletion of N_aV_1.1 during adulthood.

7. Figure 4 panels A, B, and C are a bit confusing with regard to the pie charts. The summary quantifications for static discharge patterns for wt, het, and cKO Nav1.1. mice are depicted with the amount of 'inconsistent traces' in black. This color coding is reversed for the resting discharge (absent discharge is quantified as colored; even though in wt 50% of afferents lack resting discharge, this % is higher in the mutants).

We have adjusted the color coding in this figure (now Figure 8) to improve readability. We thank the reviewer for bringing this to our attention.

8. In a set of control studies, the Nav1.1 genotypes are compared with respect to the number of proprioceptors, spindle morphology, and synaptic connectivity between MS afferents and spinal motor neurons. Interestingly, these studies appear to reveal a significant reduction in MS afferent vGlut1 positive synaptic contacts onto ChAT positive motor neurons. This is another potentially interesting result but these data appear a bit rushed. SN-MN connectivity is best compared across the same (backfilled) motor pool (and ideally by only looking at synapses from the homonymous afferents). Another aspect that may be influencing these data is the size of the synaptic boutons on the MNs. Maybe this can be resolved using higher power images (again comparing the contacts on motor neurons that innervate the same muscle)? Ultimately, the best assessment of these connections is to perform ventral root recordings after stimulating the dorsal roots. (FWIW, the DAPI panels in Figure 7 are more distractive than useful; an orientating schematic may also help).

We thank the reviewer for these comments and have re-done our analysis. Please see our comments above to Reviewer #1. We have also adjusted the figure (now Figure 7) to make the information we are trying to convey clearer.

9. There are various other studies that have performed in vitro recordings on proprioceptor afferents with sometimes divergent observations (e.g. Woo et al., 2015; Florez Paz et al., 2016; Zheng, et al., 2019; Oliver et al., 2021). It would have been interesting to see a discussion of the current observations in the context of these other studies.

We thank the Reviewer for this suggestion. The papers mentioned perform current clamp analyses where they investigate the potential contributions of voltage-gated potassium channels to proprioceptor excitability, as well as voltage-clamp recordings of mechanically gated currents. While there are differences across papers in current-clamp recordings from proprioceptors, this is due to bath temperature and is something we have directly observed as well. We have cited this work in other portions of the manuscript but find a deeper discussion of these findings to be a bit distracting from the main point of our findings.

Reviewer #3 (Recommendations for the authors):This is a very thorough study on Nav1.1 function in the peripheral nervous system. The data are generally of high quality, but I have some questions about the patch-clamp data:– Voltage-gated sodium channels are very fast gating, therefore it is crucial to have good space and voltage-clamp conditions for their measurement. Pipette resistances up to 6Mohm, as used here, are very high, and it is quite likely that the currents recorded by the authors were not well clamped. The quality of the clamp can be assessed by the shape of the current traces, thus the authors should show example IV traces, which allow judgement of the current at low voltages, where the driving force is high, but the open probability of the channels is low. This should be added to Figure 2.

We agree regarding the challenges of recording from native Na_V_ currents and thank the Reviewer for this comment. We have redone all voltage-clamp experiments in the current manuscript. We changed our recording solutions, lowering the concentration of extracellular sodium to 15 mM to ensure good voltage clamp, and used pipettes with tip resistances ranging from 3-5Mohm. We now include an IV curve in a new Figure 3, clearly showing the quality of our voltage clamp. We have updated this information in the methods section.

– Figure 2: As stated in the first point, the quality of the presented data is hard to assess. Please show the stimulation protocols. Please show a complete IV curve. At which potential was current decay assessed? Why not over a range of potentials, which would give a better understanding of the impact during an AP. Time to peak – in your case called rise time, is very fast in Navs. Again, clamping conditions are crucial. It seems that in the ICA group there is an outlier, which most likely has a higher series resistance (how high was it in general?) and thus should be removed. If done so, it is quite likely, that there will be no significant difference anymore.

In addition to the changes mentioned above, we also added stimulation protocols and have now also included information about of series resistance to the methods section, which ranged from 6-11Mohm, depending on pipette tip resistance.

– The blocker used: ICA is also effective against Nav1.3 as mentioned in the manuscript. Please show that your cultures do not express NAv1.3 or only to a minor degree. Dissociation is quite an axotomy and may thus induce Nav1.3 expression in mice. Alternatively, use cells that are unlikely to express Nav1.1 as a control for your blocker, such as MrgprD+ neurons, which you investigated in Figure 1.

As mentioned in response to Reviewer #1, we did find a small but significant effect of ICA on large-diameter sensory neurons harvested from Na_V_1.1cko animals. We have updated the manuscript accordingly and adjusted our interpretations given this caveat, though our overall conclusion is still strongly supported by our ex vivo and behavioral data.

– Discussion lines 586ff: it is easy to investigate the persistent current in your recordings. Please do so, so you can substantiate your interpretation and discussion of this point. Also, using the correct protocol it is possible to investigate if the proprioceptors show resurgent currents. An in-depth investigation of the biophysics of Nav1.1 in their expressing cells (i.e. proprioceptors) would strengthen the manuscript a lot. It could also help to understand why the channel specifically is so important for proprioceptor function, while the other channels, which are expressed in these cells, cannot compensate for it. How quickly is the Nav1.1 component recovering in proprioceptors? is this really the important characteristic which is missing in the ko mice?

We are very interested in the discrete biophysical contributions of Na_V_1.1 in proprioceptors. We have added a characterization of the whole-cell sodium current (now Figure 3) to the manuscript, including activation and inactivation curves, recovery from fast inactivation and entry into slow inactivation. Given the potential lack of specificity of ICA on native sodium channels or the upregulation of Na_V_1.3 due to culturing, we did not use ICA to determine the contribution of Na_V_1.1 to these features. As such, we also did not specifically investigate persistent or resurgent currents in the currents study. We plan to do such experiments in the Na_V_1.1^cko^ animals in future experiments as we grow this new line of investigation in the lab.

– Figure 1: please explain E and F better – what is plotted on the x-axis? What is the value 76.5 mentioned in the text in line 255?

We have added language to the results and methods section to clarify these data. Please see our response to Reviewer #2. The coefficient of variation is a measure of variability in data. In the updated manuscript, we include RNAscope analysis of Na_V_1.6 and Na_V_1.7 and compare variability in signal intensity in proprioceptors between isoforms.

– I am a bit confused by your statement in line 293 on the Vhalf of activation of Nav1.1 being -15mV. you would at least need to compare it to the Vhalf of the other Navs in order to make a statement on their involvement in the threshold. How did you determine this relatively negative threshold? In general, please provide more info on your ephys analysis (was tau obtained by single expt fitting? did you include an offset, was this changed by ICA?).

We thank the Reviewer for this suggestion. We have removed the discussion of V_1/2_ of activation from the manuscript and have now added additional information regarding the acquisition and analysis of our in vitro electrophysiological data to the Methods section to clarify the concerns raised here.

– Suppl Figure 2: which mouse is this? it is good to see the expression also in the ventral horn, please also show a 40x of the ventral horn.

The information about the mouse line can be found in the main text:

“We confirmed that our mouse model selectively targeted sensory neurons by crossing a Pirt^Cre^ driver with a fluorescent reporter line (Pirt^Cre^;Rosa26^Ai14^). We observed little-to-no neuronal expression of TdTomato in both dorsal and ventral spinal cord (Figure 5 —figure supplement 2).”

We have added a 40X ventral horn image to the figure. Thank you for this suggestion.

– Please define Lo also in the Figure legend or in the text. Figure 4: what does the percentage during the resting state refer to? fibers – time recorded?

In the methods section, Lo is defined as: “Muscles were held at optimal length (Lo), or the length of the muscle that maximal force of twitch contraction occurred.” We have also added that the Lo abbreviation means optimal length in the Figure 8 legend (fomerly Figure 4).

For the resting discharge, we have quantified the percentage of afferents that had a resting discharge of >1Hz measured before all of the 9 stretches. We have clarified this in the Figure 8 legend to read: The final pie charts represent the proportion of afferents that exhibited resting discharge at Lo for every stretch for each genotype

– Fi.5A-c: how did you test for differences in the percentage of entrainment? cKO has 64% vs fl/fl 33% – this seems to be quite a difference (in contrast to your statement in line 433 where you describe them to be similar).

This was an error on our part. For fl/fl animals 33% was the percentage of fibers did not entrain. We have adjusted this in an updated Figure 9. Additionally in our methods section we state:

“Average firing rate during the 9 s of vibration and whether the unit could entrain in a 1:1 fashion to vibration was also determined.”

– Figure 5D: what does FPS stand for?

We apologize for the error, the correct y axis label should be Firing Rate (Hz)

– Figure 8: you mention Nav1.6 and Nav1.7 in your figure. Are they expressed in your RNAscope data in proprioceptors? Do you have any other data on other Nav subtypes?

Please see previous comments above addressing this point. We have added expression and functional data for Na_V_1.6 and Na_V_1.7 to Figure 2.

– Discussion: it would be nice to mention that Nav channels are also mechanosensitive, with potential differences among sodium channel subtypes. This could contribute to their function in proprioceptors.

We thank the Reviewer for this interesting point. Currently, only Na_V_1.4 and Na_V_1.5 channels have been shown to be mechanosensitive. It is unclear if this is a general feature of Na_V_s or whether it is specific to these isoforms. Given that is in unclear whether Na_V_1.1 is mechanosensitive, we refrained from discussing how its putative mechanosensitivity could impact proprioceptor signaling, as we found it to be too speculative.